# An exposome atlas of serum reveals the risk of chronic diseases in the Chinese population

Lei You[1,2,3,8], Jing Kou[4,8], Mengdie Wang[1,3,5,8], Guoqin Ji[1,3,6,8], Xiang Li[4], Chang Su[7], Fujian Zheng[1,2,3], Mingye Zhang[4], Yuting Wang[1,2,3], Tiantian Chen[1,2,3], Ting Li[1,2,3], Lina Zhou[1,2,3], Xianzhe Shi [1,2,3], Chunxia Zhao[1,2,3], Xinyu Liu [1,2,3] ✉, Surong Mei[4] ✉ & Guowang Xu [1,2,3] ✉

Although adverse environmental exposures are considered a major cause of chronic diseases, current studies provide limited information on real-world chemical exposures and related risks. For this study, we collected serum samples from 5696 healthy people and patients, including those with 12 chronic diseases, in China and completed serum biomonitoring including 267 chemicals via gas and liquid chromatography-tandem mass spectrometry. Seventy-four highly frequently detected exposures were used for exposure characterization and risk analysis. The results show that region is the most critical factor influencing human exposure levels, followed by age. Organo-chlorine pesticides and perfluoroalkyl substances are associated with multiple chronic diseases, and some of them exceed safe ranges. Multi-exposure models reveal significant risk effects of exposure on hyperlipidemia, metabolic syndrome and hyperuricemia. Overall, this study provides a comprehensive human serum exposome atlas and disease risk information, which can guide subsequent in-depth cause-and-effect studies between environmental expo-sures and human health.

The "exposome"[1], complementing the genome, offers a promising solution for characterizing environmental chemical factors and their nonnegligible influences on chronic diseases[2]. The episome encompasses life-course environmental factors, which are generally difficult to be measured[3]. Thus, a "top-down" strategy, the "blood exposome", was proposed to directly reflect an individual's internal chemical environment by allowing measurement of all the chemicals in blood[3,4]. Known as "human biomonitoring", this method is currently being applied to monitor various chemicals in human blood, including organochlorine pesticides (OCPs)[5–7], organophosphorus pesticides

(OPPs)[8], herbicides[9], veterinary drugs[10], perfluoroalkyl substances (PFASs)[11], polycyclic aromatic hydrocarbons (PAHs)[7,12], polychlorinated biphenyls (PCBs)[5,7], and phthalates[13]. Although these targeted methods accurately monitor contaminants in human blood, each method can measure only a small number of chemicals with similar properties or structures. Indeed, the complexity of co-exposure to multiple cate-gories in the real world highlights the limitations of current methods, especially for exploring the environmental causes of disease.

In addition, the distribution characteristics of environmental chemical levels and risks in different populations have been studied to

[1]CAS Key Laboratory of Separation Science for Analytical Chemistry, Dalian Institute of Chemical Physics, Chinese Academy of Sciences, 457 Zhongshan Road, Dalian 116023, China. [2]University of Chinese Academy of Sciences, Beijing 100049, China. [3]Liaoning Province Key Laboratory of Metabolomics, Dalian 116023, China. [4]State Key Laboratory of Environment Health (Incubation), Key Laboratory of Environment and Health, Ministry of Education, Key Laboratory of Environment and Health (Wuhan), Ministry of Environmental Protection, School of Public Health, Tongji Medical College, Huazhong University of Science and Technology, #13 Hangkong Road, Wuhan, Hubei 430030, China. [5]School of Public Health, China Medical University, No. 77 Puhe Road, Shenbei New District, Shenyang 110122, China. [6]School of Life Science, China Medical University, No. 77 Puhe Road, Shenbei New District, Shenyang 110122, China. [7]National Institute for Nutrition and Health, Chinese Center for Disease Control and Prevention, Beijing 100050, China. [8]These authors contributed equally: Lei You, Jing Kou, Mengdie Wang, Guoqin Ji. ✉e-mail: liuxy2012@dicp.ac.cn; surongmei@hust.edu.cn; xugw@dicp.ac.cn

provide guidance on prevention and control policies as well as protection for susceptible individuals. Previous studies have shown that epidemiological factors, such as age[6,7,14–16], sex[6,17,18], and education and income levels[6,15,19], may influence the level of chemical residues in humans, and different distributions of serum chemical levels are associated with epidemiological factors. Furthermore, the same exposure may present a significantly greater risk effect in specific populations[20], and sex-specific exposure-risk relationships are of particular interest[21–23]. Nevertheless, existing research has not yet elucidated the relationship between chemical exposure and population diversity, especially in large-scale general population cohorts.

Chronic diseases are indisputably the predominant challenge to health globally[24]. Both genetic and environmental factors contribute to multiple chronic diseases, with the latter being more important[25]. Currently, a high-profile issue is the associations between chemical exposures and disease outcomes. Environmental scientists and epidemiologists have widely studied these associations in multiple chronic diseases, such as hypertension[23,26], diabetes[27,28], hyperlipidemia[29–31], and hyperuricemia[32,33], among others. However, existing association studies are not sufficiently robust and are sometimes even contradictory, which is probably due to limited sample sizes and large individual differences in exposures. Moreover, the health effects of a single chemical have been the focus of most cohort studies, whereas the "cocktail effects" caused by exposure mixtures have largely been neglected, especially at real-world environmental concentrations.

In this work, we aimed to comprehensively characterize the human serum exposome, provide an exposure characteristic atlas resource, and define the exposure-disease risk relationship. To this end, we first enrolled a large-scale general population of 5696 people and collected serum samples, basic epidemiological information and chronic disease-related clinical parameters. Then, two targeted methods covering 267 environmental chemicals typically reported were established to quantify a portion of the serum exposome: one based on the gas chromatography–tandem mass spectrometry (GC–MS/MS) platform for quantification of 97 chemicals and another based on the liquid chromatography (LC)–MS/MS platform for quantification of 170 chemicals. Next, we assessed levels and the risk of serum exposures in different people stratified by epidemiological information. Finally, single-exposure and multi-exposure models were used to define correlations between chronic diseases and the serum exposome and to reveal key risk factors.

## Results

To assess the risk of chronic disease from the serum exposome, (i) we designed a cohort of 5696 healthy and chronic disease patients from 15 provinces in China; the 12 chronic diseases included diabetes, hyperuricemia, obesity, hypercholesterolemia, hypertriglyceridemia, metabolic syndrome, high diastolic blood pressure, high systolic blood pressure, abdominal obesity, hypertension, high low-density lipoprotein cholesterol and hyperlipidemia. Additionally, we collected data on 9 basic epidemiological factors and 9 clinical parameters of chronic diseases and serum samples (Table 1). A portion of the human serum exposome, which included 267 chemicals, was comprehensively characterized via GC–MS/MS and LC–MS/MS; 74 chemicals were found at high frequencies and further studied as key targets of interest (Fig. 1a, Table 2). (ii) The participants were grouped using basic epidemiological information, after which residual levels of chemicals in serum were determined in the stratified population (Fig. 1b). (iii) Robust associations between exposures and risk of chronic disease were established using single-exposure and multi-exposure models together to specify chemical residues at risk for chronic diseases (Fig. 1c).

### Determination of serum chemicals and batches

Human biomonitoring is prioritized for chemicals that possibly accumulate in the body and cause health effects based on the literature and public databases. A total of 97 and 170 chemicals, including OCPs, OPPs, herbicides, insecticides, fungicides, veterinary drugs, food additives, PAHs, PCBs, PFASs, and phthalates (Supplementary Fig. 1a), were selected as priority lists and monitored by GC–MS/MS and LC–MS/MS; specific information is given in Supplementary Data 1. For 74 exposures, 50% higher detection frequency in serum samples was found (Table 2, Supplementary Fig. 1b). The detection frequencies and concentration levels of all the exposures are given in Supplementary Data 2.

In general, routine maintenance of instrumentation is necessary and performed regularly to ensure the quality of the instrument and the repeatability and stability of the data during long-term large-scale sample analysis, and batches were generated accordingly. The composition of each batch is shown in Supplementary Fig. 2, which shows the running sequences of the calibration curves, quality control (QC)

## Table 1 | Epidemiological information and chronic disease parameters of the samples involved in this study

| Characteristics | Total | Control | Diseases |
|---|---|---|---|
| gender (M/F) | 2607/3089 | 906/1235 | 1701/1854 |
| age (years) | 51 ± 17 | 44 ± 19 | 56 ± 13 |
| cigarette smoking history (yes/no) | 1396/4300 | 436/1705 | 960/2595 |
| alcohol drinking history (yes/no) | 1564/4132 | 487/1654 | 1077/2478 |
| education level | | | |
| primary school, N (%) | 2108 (100) | 774 (37) | 1334 (63) |
| junior high school, N (%) | 1828 (100) | 688 (38) | 1140 (62) |
| high school, N (%) | 1236 (100) | 432 (35) | 804 (65) |
| university and above, N (%) | 524 (100) | 247 (47) | 277 (53) |
| marital status | | | |
| married, N (%) | 4853 (100) | 1689 (38) | 3164 (65) |
| unmarried, N (%) | 468 (100) | 367 (78) | 101 (22) |
| others, N (%) | 375 (100) | 85 (23) | 290 (77) |
| income (RMB) | 22,211 ± 37,160 | 21,536 ± 35,618 | 22,617 ± 38,058 |
| region (north/south) | 2116/3580 | 604/1537 | 1411/2144 |
| sampling No. in Aug./Sep./Oct./Nov. | 1473/1790/2114/319 | 403/765/853/120 | 1070/1025/1261/199 |
| uric acid (μmol/L) | | | |
| male | 323 ± 85 | 284 ± 65 | 343 ± 87 |
| female | 265 ± 74 | 238 ± 57 | 282 ± 80 |
| glycated hemoglobin (%) | 5.7 ± 0.9 | 5.4 ± 0.4 | 5.9 ± 1.0 |
| LDL-C[a] (mmol/L) | 3.1 ± 0.9 | 2.5 ± 0.5 | 3.5 ± 0.9 |
| triglycerides (mmol/L) | 1.4 ± 1.0 | 1.0 ± 0.4 | 1.7 ± 1.2 |
| total cholesterol (mmol/L) | 4.9 ± 1.1 | 4.3 ± 0.6 | 5.3 ± 1.1 |
| systolic blood pressure (mmHg) | 127 ± 20 | 115 ± 13 | 134 ± 19 |
| diastolic blood pressure (mmHg) | 81 ± 12 | 74 ± 9 | 85 ± 11 |
| waistline (cm) | | | |
| male | 85 ± 12 | 79 ± 11 | 88 ± 11 |
| female | 82 ± 11 | 77 ± 10 | 86 ± 10 |
| BMI (kg/m²) | 23.9 ± 3.8 | 21.9 ± 3.2 | 25.1 ± 3.7 |

[a]LDL-C, low density lipoprotein cholesterol.

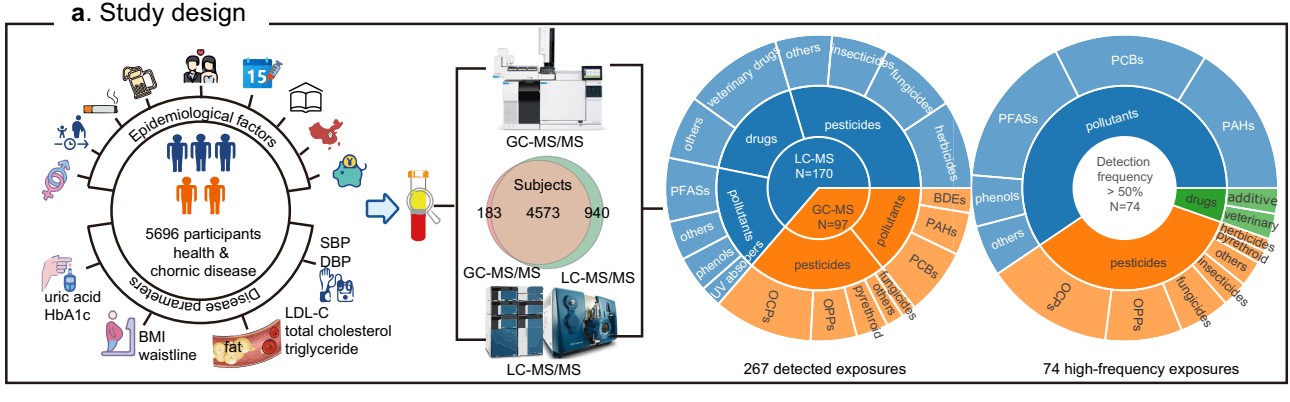

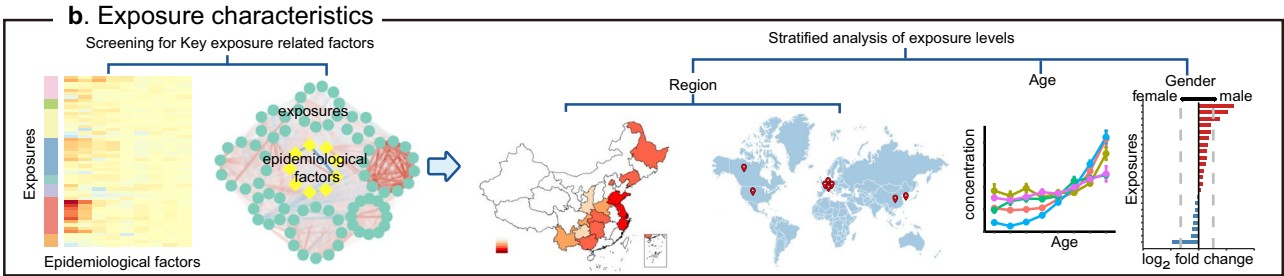

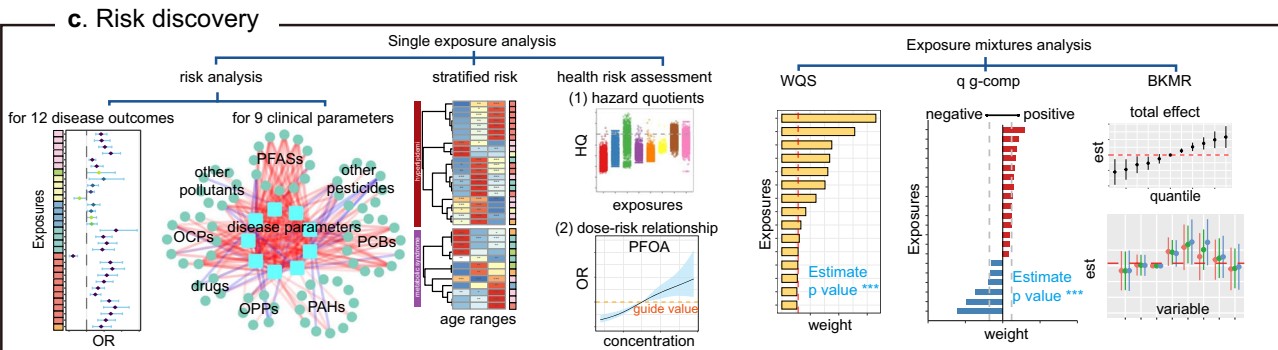

**Fig. 1 | Overview of the study. a** Study design: Serum samples of 5696 participants were included in the study, 9 basic epidemiological information and 9 chronic disease-related clinical parameters were collected. Two platforms of LC-MS/MS and GC-MS/MS were used to detect 267 exposures, and 74 high-frequency exposures were determined. **b** Exposures characteristics: Exposures and basic epidemiological factors were associated, according to these factors, participants were stratified to analyze the distribution characteristics of exposures. **c** Risk discovery. First, single-exposure analysis showed the risk, stratified risk and health risk assessment of each exposure to chronic disease, and then exposure mixtures analysis showed risk effects of exposure mixtures on related chronic diseases based on 3 multi-exposure models. HbA1c glycated hemoglobin, BMI body mass index, LDL-C low density lipoprotein cholesterol, SBP systolic blood pressure, DBP diastolic blood pressure, OCP organochlorine pesticide, OPP organophosphorus pesticide, PAH polycyclic aromatic hydrocarbon, PCB polychlorinated biphenyl, PFAS per-fluoroalkyl substance, WQS weighted quantile sum regression, q g-comp quantile g-computation, BKMR Bayesian kernel machine regression.

samples, and actual samples. A calibration curve was constructed at the beginning of each batch. The accuracy of each calibration curve was evaluated at low, medium, and high concentrations. Specifically, 2, 5, and 20 ng/mL spiked concentrations were used for accurate evaluation of 29 calibration curves using the GC–MS/MS platform. Spiked concentrations of 1, 10, and 100 ng/mL were used for evaluating the accuracy of 20 calibration curves using the LC–MS/MS platform. In total, 49 independent calibration curves were constructed during the whole sample analysis process for batch-specific quantification, and the results revealed good accuracy at low, medium, and high concentrations (Supplementary Fig. 3, Supplementary Data 3, 4). QC samples were used to evaluate batch effects and the stability and accuracy of the data. Batch effects were observed in the raw signal (Supplementary Fig. 4a, b) but were greatly reduced after batch-specific calibration curve quantification and correction for multiple internal standards (Supplementary Fig. 4c–f). Ninety-two percent of the 267 detected exposures and 84% of the 74 high-frequency exposures met the requirement of relative standard deviations (RSDs) less

than 30% in the QC samples, indicating good stability of the data (Supplementary Fig. 5a, b; Supplementary Data 5, 6). In addition, 98% and 79% of the exposures monitored by the GC–MS/MS and LC–MS/MS platforms, respectively, met the requirement of accuracy between 80% and 120% in the QC samples, indicating good data accuracy (Supplementary Fig. 5c, d). Finally, the samples were randomly analyzed using both platforms according to a given disease to achieve sample randomization (Supplementary Data 7).

## Correlation between basic epidemiological factors and serum exposures
Considering that contact and accumulation of chemicals vary widely across populations, specific relationships between serum exposure and epidemiological factors, such as region, sex, age, other social factors and lifestyle, were examined in this study. First, the influence of each epidemiological factor on serum exposure was determined, samples were stratified according to each epidemiological factor (Supplementary Fig. 6), and serum exposures associated with

**Table 2 | Concentration levels of 74 high-frequency exposures in human serum**

| Compound | Detection frequency (%) | | Concentration (ng/mL) | |
| --- | --- | --- | --- | --- |
| | Total | Highest in 15 provinces | Geometric mean (95% CI) | Median |
| alpha-hexachlorocyclohexane (HCH) | 19 | 57 | 0.0201 (0.0195, 0.0206) | <LOQ |
| beta-HCH | 89 | 100 | 0.313 (0.297, 0.330) | 0.306 |
| p,p'-dichlorodiphenyldichloroethane (DDD) | 37 | 88 | 0.0074 (0.0072, 0.0077) | <LOQ |
| p,p'-dichlorodiphenyldichloroethylene (DDE) | 98 | 100 | 1.91 (1.82, 2.00) | 2.210 |
| p,p'-dichlorodiphenyltrichloroethane (DDT) | 62 | 83 | 0.0451 (0.0436, 0.0467) | 0.037 |
| methoxychlor | 42 | 60 | 0.0273 (0.0266, 0.0281) | <LOQ |
| pentachlorobenzene | 26 | 53 | 0.0025 (0.0023, 0.0027) | <LOQ |
| hexachlorobenzene (HCB) | 87 | 100 | 0.308 (0.289, 0.328) | 0.515 |
| pentachlorophenol | 68 | 94 | 0.497 (0.471, 0.525) | 0.410 |
| mirex | 45 | 82 | 0.0176 (0.017, 0.0183) | <LOQ |
| dichlorvos | 39 | 84 | 0.0272 (0.0258, 0.0287) | <LOQ |
| dimethoate | 18 | 64 | 0.0545 (0.0529, 0.0563) | <LOQ |
| diazinon | 23 | 51 | 0.0343 (0.0333, 0.0354) | <LOQ |
| chlorpyrifos | 74 | 99 | 0.054 (0.0522, 0.0558) | 0.035 |
| triazophos | 19 | 59 | 0.0907 (0.0892, 0.0923) | <LOQ |
| mecarbam | 10 | 69 | 0.428 (0.420, 0.436) | <LOQ |
| 2,4-dichlorophenoxyacetic acid (2,4-D) | 37 | 59 | 0.0537 (0.0528, 0.0547) | <LOQ |
| iprodione | 53 | 79 | 0.0353 (0.0341, 0.0365) | 0.023 |
| triclocarban | 56 | 83 | 0.0185 (0.018, 0.019) | 0.014 |
| chlorothalonil | 41 | 86 | 0.115 (0.110, 0.120) | <LOQ |
| triclosan | 55 | 92 | 0.0515 (0.0498, 0.0532) | 0.033 |
| lufenuron | 26 | 84 | 0.0919 (0.0907, 0.0932) | <LOQ |
| fipronil sulfone | 100 | 100 | 0.278 (0.269, 0.288) | 0.271 |
| etofenprox | 73 | 93 | 0.167 (0.158, 0.176) | 0.174 |
| isoprocarb | 43 | 84 | 0.141 (0.137, 0.145) | <LOQ |
| Indole-3-butyric acid (IBA) | 95 | 99 | 2.14 (2.10, 2.19) | 2.190 |
| cortisone | 99 | 100 | 14.56 (14.34, 14.78) | 15.880 |
| cortisol | 98 | 100 | 83.01 (81.40, 84.64) | 93.000 |
| cyclamic acid | 90 | 96 | 0.762 (0.714, 0.814) | 0.679 |
| acesulfame | 65 | 88 | 0.346 (0.325, 0.369) | 0.170 |
| acenaphthylene | 37 | 59 | 0.0086 (0.0078, 0.0095) | <LOQ |
| fluorene | 48 | 75 | 0.0357 (0.0316, 0.0402) | <LOQ |
| anthracene | 44 | 63 | 0.0336 (0.0311, 0.0364) | <LOQ |
| acenaphthene | 39 | 100 | 0.022 (0.0203, 0.0238) | <LOQ |
| phenanthrene | 36 | 100 | 0.111 (0.103, 0.119) | <LOQ |
| benzo (a) anthracene | 100 | 93 | 0.0185 (0.0178, 0.0191) | 0.022 |
| chrysene | 82 | 98 | 0.0134 (0.0128, 0.014) | 0.101 |
| fluoranthene | 32 | 91 | 0.0146 (0.0138, 0.0153) | <LOQ |
| pyrene | 97 | 100 | 0.037 (0.0334, 0.0409) | 1.010 |
| benzo[b]fluoranthene | 69 | 93 | 0.0202 (0.0186, 0.0219) | 0.0089 |
| benzo[k]fluoranthene | 47 | 70 | 0.0123 (0.0116, 0.013) | <LOQ |
| benzo[a]pyrene | 55 | 88 | 0.147 (0.135, 0.160) | 0.0077 |
| PCB-77 | 8 | 54 | 0.0078 (0.0077, 0.0079) | <LOQ |
| PCB-105 | 13 | 56 | 0.0042 (0.0041, 0.0043) | <LOQ |
| PCB-114 | 13 | 56 | 0.0081 (0.008, 0.0082) | <LOQ |
| PCB-118 | 24 | 51 | 0.0096 (0.0094, 0.0098) | <LOQ |
| PCB-126 | 24 | 64 | 0.0052 (0.0051, 0.0054) | <LOQ |
| PCB-138 | 50 | 74 | 0.0095 (0.0092, 0.0099) | 0.0052 |
| PCB-153 | 48 | 79 | 0.0092 (0.0089, 0.0095) | <LOQ |
| PCB-156 | 37 | 58 | 0.0111 (0.0109, 0.0113) | <LOQ |
| PCB-167 | 11 | 56 | 0.0081 (0.008, 0.0082) | <LOQ |
| PCB-169 | 19 | 55 | 0.005 (0.0049, 0.0052) | <LOQ |
| PCB-180 | 32 | 59 | 0.0113 (0.011, 0.0116) | <LOQ |
| PCB-183 | 17 | 52 | 0.0086 (0.0084, 0.0087) | <LOQ |

**Table 2 (continued) | Concentration levels of 74 high-frequency exposures in human serum**

| Compound | Detection frequency (%) | | Concentration (ng/mL) | |
|---|---|---|---|---|
| | Total | Highest in 15 provinces | Geometric mean (95% CI) | Median |
| perfluoro-n-pentanoic acid (PFPeA) | 98 | 99 | 1.10 (1.07, 1.12) | 1.210 |
| perfluorooctanoic acid (PFOA) | 100 | 100 | 3.99 (3.87, 4.12) | 3.260 |
| perfluorononanoic acid (PFNA) | 98 | 100 | 0.813 (0.786, 0.84) | 0.873 |
| perfluorodecanoic acid (PFDA) | 98 | 100 | 0.723 (0.698, 0.75) | 0.737 |
| perfluoroundecanoic acid (PFUnDA) | 97 | 100 | 0.514 (0.496, 0.532) | 0.561 |
| perfluorododecanoic acid (PFDoDA) | 83 | 98 | 0.0512 (0.049, 0.0534) | 0.068 |
| perfluorotridecanoic acid (PFTrDA) | 83 | 98 | 0.0789 (0.0759, 0.0821) | 0.095 |
| perfluorohexanesulfonate (PFHxS) | 90 | 100 | 0.387 (0.373, 0.401) | 0.406 |
| perfluoroheptanesulfonic acid (PFHpS) | 99 | 100 | 0.102 (0.0996, 0.105) | 0.102 |
| perfluorooctanesulfonate (PFOS) | 100 | 100 | 4.65 (4.51, 4.80) | 4.890 |
| 6:2 chlorinated polyfluoroalkyl ether sulfonate (6:2 Cl-PFAES) | 83 | 100 | 2.57 (2.47, 2.67) | 2.360 |
| bis[2-(perfluorohexyl)ethyl] phosphate (6:2 diPAP) | 22 | 50 | 2.22 (2.19, 2.25) | <LOQ |
| bisphenol F | 16 | 53 | 0.409 (0.405, 0.413) | <LOQ |
| bisphenol A | 41 | 66 | 1.10 (1.08, 1.12) | <LOQ |
| bisphenol S | 12 | 63 | 0.195 (0.193, 0.197) | <LOQ |
| bisphenol AF | 17 | 59 | 0.397 (0.394, 0.40) | <LOQ |
| triphenyl phosphate (TPHP) | 43 | 71 | 0.518 (0.511, 0.526) | <LOQ |
| monocyclohexyl phthalate (MCHP) | 52 | 60 | 0.147 (0.144, 0.15) | 0.113 |
| monoethyl phthalate (MEP) | 93 | 99 | 2.56 (2.50, 2.62) | 2.880 |
| dibutyl phosphate (DBP) | 28 | 55 | 0.481 (0.474, 0.488) | <LOQ |

*LOQ* limit of quantification.

epidemiological factors were identified. Region was the major factor influencing the level of exposure in human serum, which explained 15.6% of the variance in exposure (Fig. 2a). The chemicals most influenced by regional factors were PFASs, especially perfluorocarboxylic acids, which included perfluorooctanoic acid (PFOA), perfluorononanoic acid (PFNA), perfluorotridecanoic acid, and perfluorodecanoic acid (PFDA) (Fig. 2b). Sampling time was found to be the second most important factor. However, sampling time and region exhibited similar associations with serum exposure (Fig. 2b), which was probably because the sampling was conducted region by region over a certain period (Supplementary Fig. 6e). Furthermore, as the sampling time was mainly concentrated across only 3 months, the effect of sampling time on the serum exposome was considered attributable to regional factors without additional discussion. The next important factor causing variation in human serum exposure was age, which accounted for 1.4% of the variation in exposure. Factors other than region and age had relatively limited effects on exposure levels (Fig. 2a, b). Principal component analysis revealed significant separation trends among samples from people of different regions and ages based on human chemical residues, but no such trends were observed for other factors (Fig. 2c, d; Supplementary Fig. 7). Although human serum exposures were significantly associated with epidemiological factors, they correlated much less than chemicals in the same category. The strongest association was demonstrated among PFASs, followed by OCPs, PCBs and PAHs (Fig. 2e; Supplementary Fig. 8). This suggests that chemicals in the same category are likely from similar sources of contamination.

**Distribution characteristics of human serum exposures based on region, age and other factors**
Considering that region was the most important factor influencing human exposure levels, the population was stratified according to region to explore the distribution characteristics of exposures. Overall, the highest concentrations of total human serum chemicals were found in Shanghai, followed by Zhejiang, Jiangsu and Shandong, which are located in eastern coastal areas and have dense populations and

well-developed industries. The lowest concentrations of chemicals were found in Shaanxi and Guizhou, which are inland areas with lower populations and fewer industries (Fig. 3a). Among the various types of human serum exposure, the concentration levels of drugs were highest, followed by PFASs, which varied greatly among provinces (Fig. 3b). People in the Yangtze River Delta had higher serum PFASs, especially in Shanghai and Jiangsu (Supplementary Fig. 9a). Exposure levels of PAHs, PCBs and OCPs also differed greatly among the populations in different regions. For example, serum PAH concentrations in people from Henan, Zhejiang, and Shandong were significantly greater than those in people from other provinces (Supplementary Fig. 9b). Serum levels of PCBs and OCPs were significantly greater in Chongqing, Shanghai, and Jiangsu (Supplementary Fig. 9c, d). For OPPs, the highest exposure levels were found in the population of Chongqing (Supplementary Fig. 9e). Veterinary drugs and food additives had little difference in populations across provinces (Supplementary Fig. 9f).

To better understand exposure levels in human serum on a larger regional scale, the exposure information of some high-frequency chemicals was compared with that of seven other countries based on the literature. In comparison with other regions of the world, serum from people who live in China has the highest residue levels of p,p'-dichlorodiphenyldichloroethylene (DDE), beta-hexachlorocyclohexane (HCH), p,p'-dichlorodiphenyltrichloroethane (DDT), PFOA, perfluoroundecanoic acid and perfluoro-n-pentanoic acid (PFHpS). The highest residue levels of hexachlorobenzene (HCB), perfluorooctanesulfonate (PFOS) and PFDA were found in the serum of people in Korea. The highest residue levels of PFNA and perfluorohexanesulfonate (PFHxS) were found in the serum of people in the United States and Canada, respectively (Supplementary Fig. 10, Supplementary Data 8).

Age is an important factor that explains differences in human exposome and may be related to differences in the exposure times and metabolic levels of people of different age ranges. Therefore, exposure levels were analyzed for each age group (Fig. 3c–e). Accumulation of the majority of chemicals, such as OCPs and PFASs, in human serum increased with age (Fig. 3c, d). Among them, beta-HCH, p,p'-DDE, pyrene, and indole-3-butyric acid (IBA) increased with age, and the

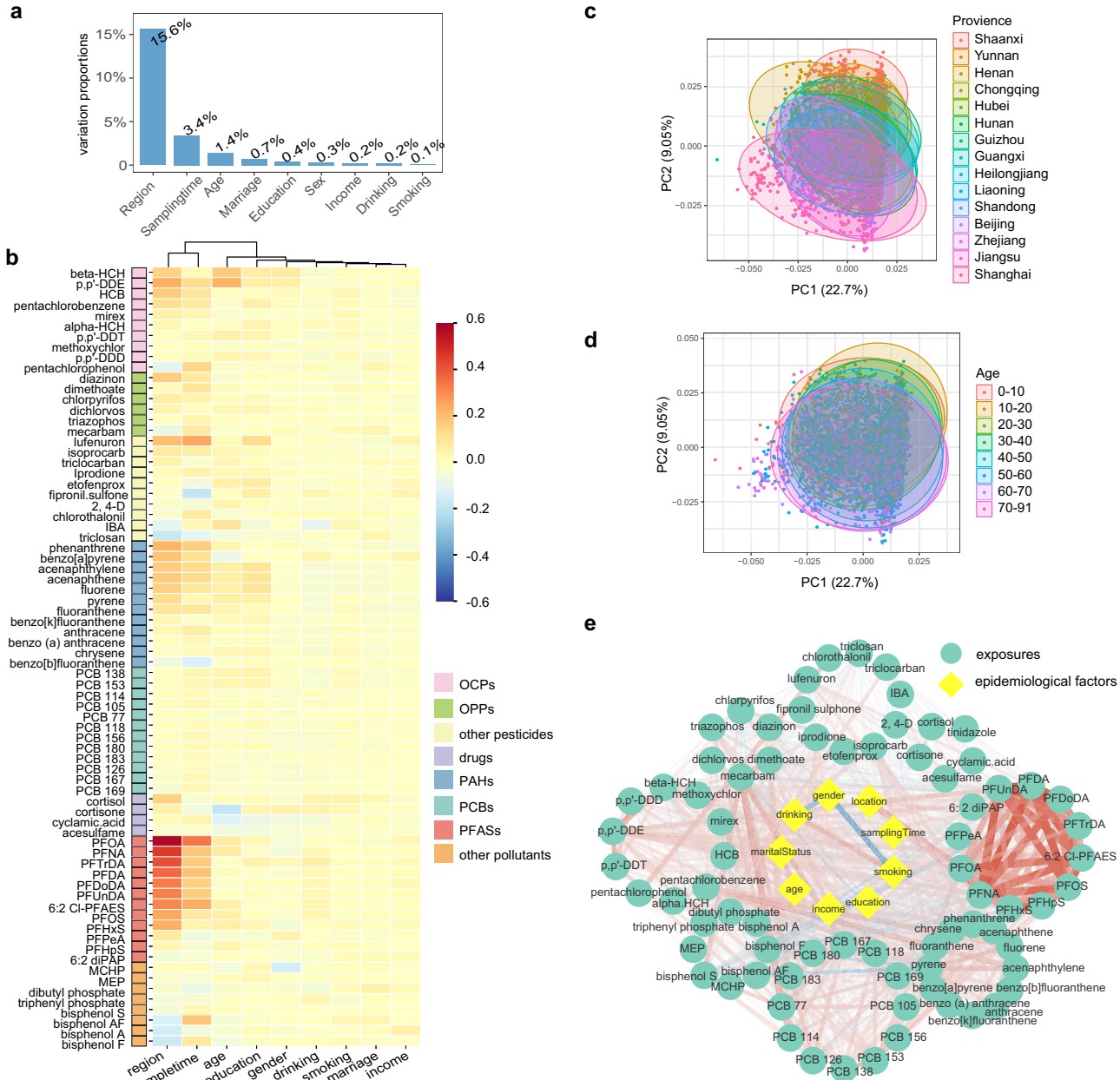

**Fig. 2 | Correlation analysis of the basic epidemiological factors and exposures.** **a** Explanation of exposure variations using epidemiological information based on variation partitioning analysis. **b** Correlations between each exposure and 9 basic epidemiological factors. The correlation coefficients obtained from the partial spearman correlation analysis were used to plot the heatmap. Principal component analysis score plot for 15 provinces (**c**) and different age ranges (**d**). **e** Correlation network of exposures and epidemiological factors. Red line represents positive correlation, blue line represents negative correlation obtained by Spearman correlation analysis. HCH hexachlorocyclohexane, DDD dichlorodiphenyldichloroethane, DDE dichlorodiphenyldichloroethylene, DDT dichlorodiphenyltrichloroethane, IBA indole-3-butyric acid, PFOA perfluorooctanoic acid, PFNA perfluorononanoic acid, PFDA perfluorodecanoic acid, PFUnDA perfluoroundecanoic acid, PFDoDA perfluorododecanoic acid, PFTrDA perfluorotridecanoic acid, PFOS perfluorooctanesulfonate, PFHxS perfluorohexanesulfonate, 6:2 Cl-PFAES 6:2 chlorinated polyfluoroalkyl ether sulfonate, 6:2 diPAP bis[2-(perfluorohexyl)ethyl] phosphate, PFPeA perfluoro-n-pentanoic acid, PFHpS perfluoroheptanesulfonic acid, MCHP monocyclohexyl phthalate, MEP monoethyl phthalate.

highest concentrations were found in people older than 70 years. The PFAS concentration increased with age until the age of 50 years (Fig. 3d). In contrast, the concentrations of some chemical agents decreased with age, with the serum of children aged less than 10 years having the highest residual levels of cyclamic acid and acesulfame (Fig. 3e), which prompted us to investigate the intake of sugar substitutes in these children.

Other factors also have an impact on serum exposure. Stratification of the population according to education and income levels revealed that most chemicals tended to increase as education and income levels increased. Specifically, these included PAHs, OCPs, and PCBs (Fig. 3f–h, Supplementary Fig. 11a). Only IBA residues showed a decreasing trend (Supplementary Fig. 11b, c). In addition to education and income, sex is an important factor influencing accumulation and excretion of chemicals in the human body. Significantly higher levels of OCPs were found in females than in males, whereas significantly lower levels of PFASs and phthalates were found in females (Fig. 3i). The chemicals most influenced by sex were beta-HCH, monocyclohexyl phthalate (MCHP) and PFHxS (Supplementary Fig. 11d–f). Finally, with regard to smoking and alcohol consumption, only male

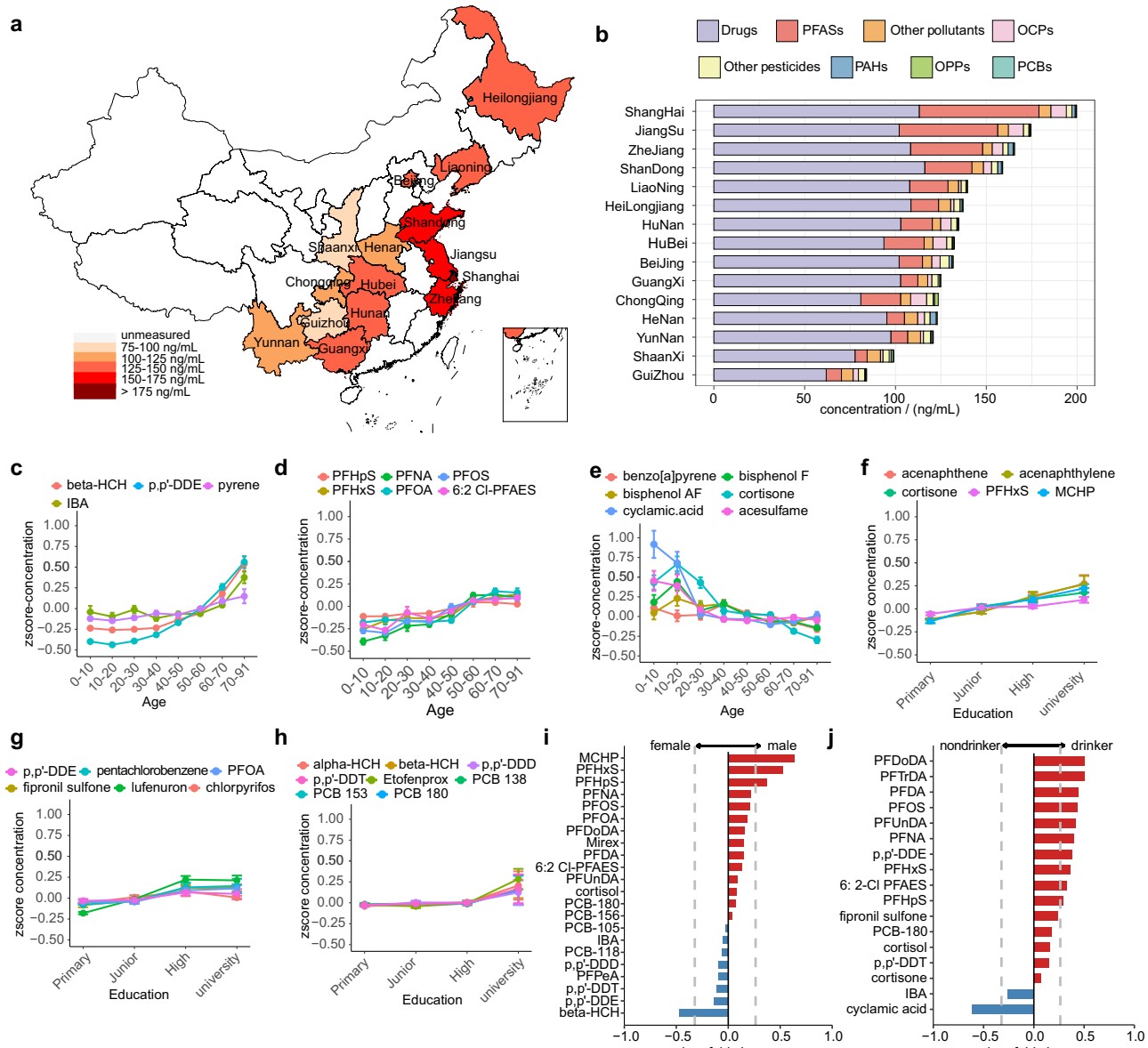

**Fig. 3 | Hierarchical analysis of exposure levels based on key epidemiological factors. a** The location of province included in this study and their total concentration of exposures. **b** Regional distribution of different categories of exposures depicted by stacked bar plot. Exposures that significantly increase (**c**, **d**) and decrease (**e**) with age. Exposures that significantly increase (**f**–**h**) with education. For figures (**c**–**h**), the concentrations scaled by the z-score method were used and error bars represent standard error of mean ($n = 5696$ biologically independent samples). Exposures that significantly change with gender (**i**) and drinking history in male (**j**). The geometric means of the exposures were used for figures (**a**, **b**, **i**, **j**). Gray dash lines of figures (**i**, **j**) represent fold change less than 0.8 and more than 1.3.

individuals were considered because women in this study rarely smoked or drank alcohol (Supplementary Fig. 6d, f). For many chemicals, significantly greater concentrations were detected in drinkers than in nondrinkers (Fig. 3j); the chemicals most influenced by drinking were cyclamic acid and six PFASs, with higher levels of the latter in drinkers (Supplementary Fig. 11g–m). Thus, the risk of exposure associated with alcohol consumption should be considered. Unlike drinking, exposure to various chemicals did not significantly differ between smokers and nonsmokers (Supplementary Fig. 11n).

## Risk analysis of chronic diseases associated with a single exposure

Whether exposure is associated with the risk of chronic disease and which chemicals are key risk factors are two questions of wide interest. In this study, 9 clinical parameters of chronic disease were subdivided into 12 related chronic disease outcomes, and the health risk of

exposure was analyzed for each outcome (Supplementary Fig. 12a, b). Among the 12 chronic diseases, three were determined by multiple clinical parameters: hyperlipidemia, metabolic syndrome, and hypertension (Supplementary Fig. 12c–e). The remaining 9 chronic diseases were determined by single clinical parameters. Well-matched controls were selected for each disease outcome (Supplementary Data 9, 10). The specific analysis scheme is shown in Supplementary Fig. 13a. For 12 disease outcomes, multiple exposure-disease risk associations were observed. Specifically, OCPs and PCBs were associated with an increased risk of hypertension, diabetes, metabolic syndrome and obesity. PFASs were associated with an increased risk of hyperlipidemia, metabolic syndrome, diabetes, and hyperuricemia (Fig. 4, Supplementary Fig. 13b). The disease associated with the greatest number of chemicals was hyperlipidemia, as was the related disease (Fig. 4a–d), followed by metabolic syndrome (Fig. 4e), demonstrating that these diseases are most affected by environmental chemicals. Most

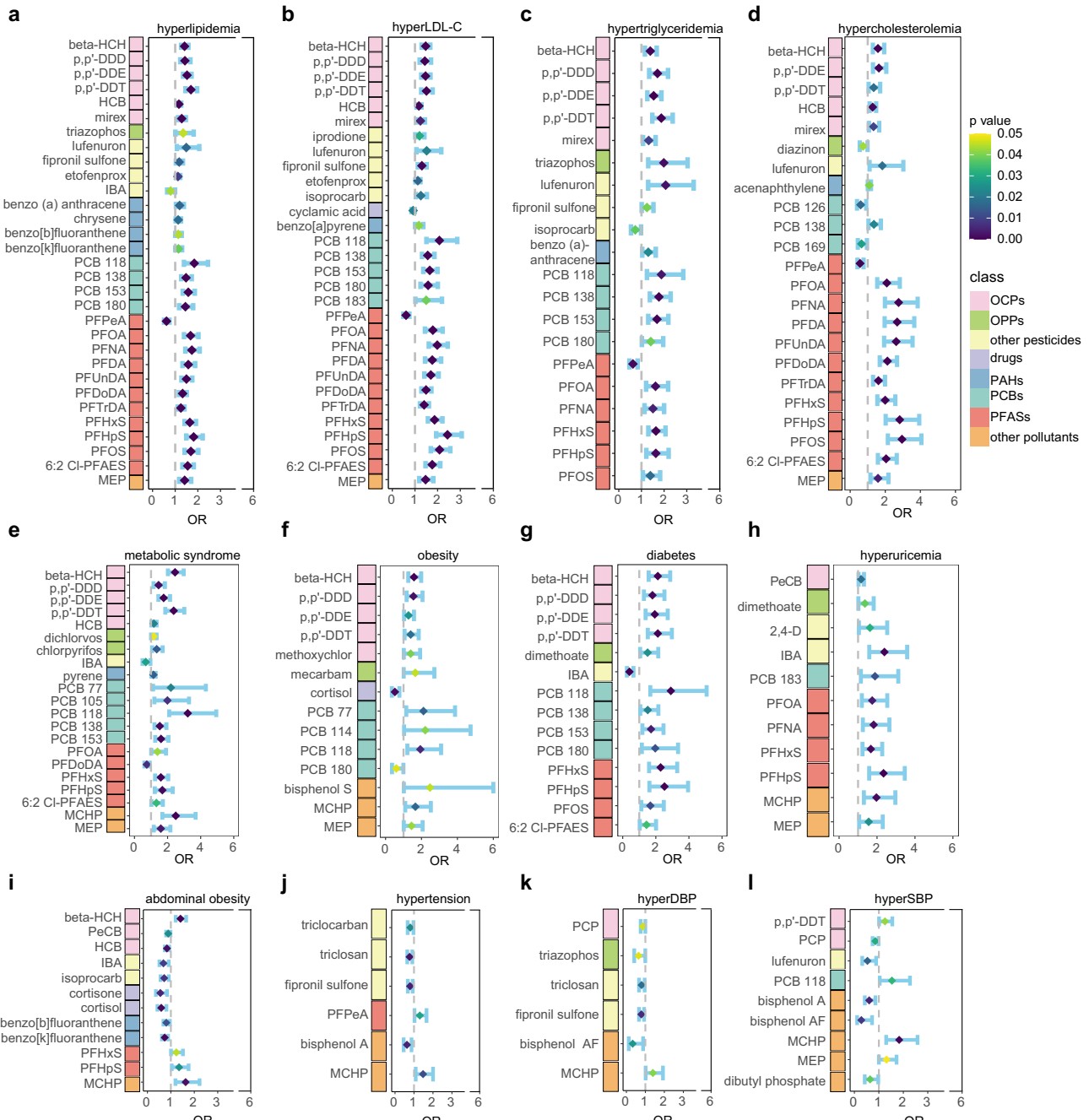

**Fig. 4 | The odds ratios of exposure to chronic diseases.** Exposures with a significant risk for hyperlipidemia (**a**), hyper low density lipoprotein cholesterol (**b**), hypercholesterolemia (**c**), hypertriglyceridemia (**d**), metabolic syndrome(**e**), obesity (**f**), diabetes (**g**), hyperuricemia (**h**), abdominal obesity (**i**), hypertension (**j**), hyper diastolic blood pressure (**k**) and hyper systolic blood pressure (**l**). The concentrations were log10 transformed so ORs represent odds ratios per one-unit increase in log-transformed exposure levels. Binary logistic regression models adjusted for age, gender, region, sampling time, education and income levels, marital status, smoking and drinking history. The position and color of diamond represent ORs and significant (two-sided, *n* = 5696 biologically independent samples), respectively. Error bars represent 95% confidence interval of ORs.

chemicals have positive effects on the risk of developing chronic diseases, except for hypertension and abdominal obesity. In particular, all chemicals had significant effects on the risk of hyperuricemia (Fig. 4h). Among all chronic disease outcomes, hypertension, and related diseases had the weakest risk relationship with chemicals (Fig. 4j–l). These results were still similar after additional adjustment for confounders of external environmental factors, including air pollution and meteorological conditions (Supplementary Fig. 14). Associations between 193 low-frequency exposures and chronic diseases were also investigated. Most chemicals were associated with an increased risk of chronic diseases, particularly metabolic syndrome, obesity, diabetes, and hyperuricemia (Supplementary Figs. 15, 16). However, compared to the results of high-frequency exposures, the significance of associations between low-frequency chemicals and chronic diseases was weaker, with larger confidence intervals, which highlights the importance of cautious interpretation of these associations. Therefore, in subsequent analyses, we focused solely on association analysis of highly frequently detected chemicals. Similar exposure risk associations were found in analysis of nine continuous clinical parameters, complementing the results of the classified outcomes described above. Thirteen chemicals

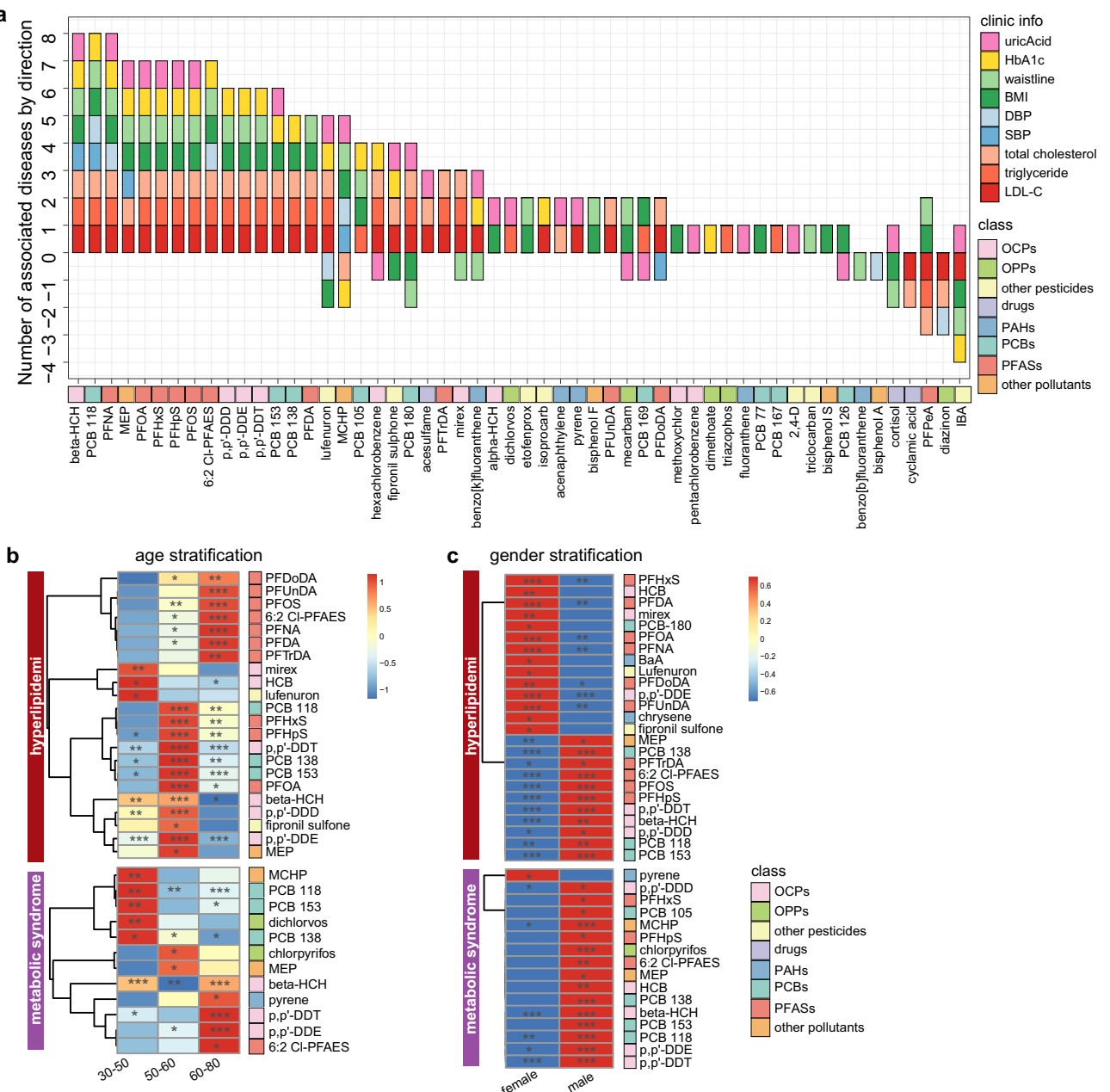

**Fig. 5 | The risk of single exposure to major chronic diseases. a** Relationship between each exposure and 9 clinical disease parameters. Included exposures were significantly associated with at least one disease outcome. Risk of exposures for hyperlipidemia and metabolic syndrome stratified by age (**b**) and gender (**c**). Scaled odds ratios were used in the heat maps of stratified risk, and * represent significant

$0.01 < p < 0.05$, ** represent significant $0.001 < p < 0.01$, *** represent significant $p < 0.001$ (two-sided). Exposures with significant associations for the corresponding diseases were used for this plot based on multiple linear regression and binary logistic regression models. All of regression models adjusted for age, gender, region, sampling time, education and income levels, marital status, smoking and drinking history.

had risk effects on five or more disease outcomes according to both regression models (Fig. 5a; Supplementary Fig. 13b). The predominant adverse effects were mainly OCPs, PFASs, and PCBs, demonstrating the nonspecific risk of these chemicals for multiple chronic diseases (Supplementary Fig. 17).

Furthermore, susceptibility to chronic diseases was investigated in populations of different ages and sexes, which can help to identify and protect susceptible patient subgroups. Hyperlipidemia and metabolic syndrome were chosen as target diseases because of the sufficient sample size and the exposure-disease associations described above. Susceptibility to exposure-induced chronic diseases was explored in three age ranges, 30-50, 50–60, and 60–80 years, representing young-aged, middle-aged, and elderly groups, respectively. Compared to

those in the middle-aged group, the elderly group presented a greater risk of exposure to hyperlipidemia; the risk factors were mainly PFASs and OCPs, which also had a stronger exposure risk for metabolic syndrome in elderly group (Fig. 5b). Given that many disease risk factors are sex-specific, we also analyzed the sex association between exposure and disease risk; no significant sex differences were found in terms of exposure risk to hyperlipidemia, but almost all men were at increased risk for metabolic syndrome (Fig. 5c). Specific differences in odds ratios (ORs) between men and women are shown in Supplementary Fig. 18.

Finally, health risk assessments of all monitored individuals were performed using reported exposure guidance values. Available reference dose (RfD) values of 11 PFASs were found, and available exposure guidance values for blood (biomonitoring equivalent (BE), Human

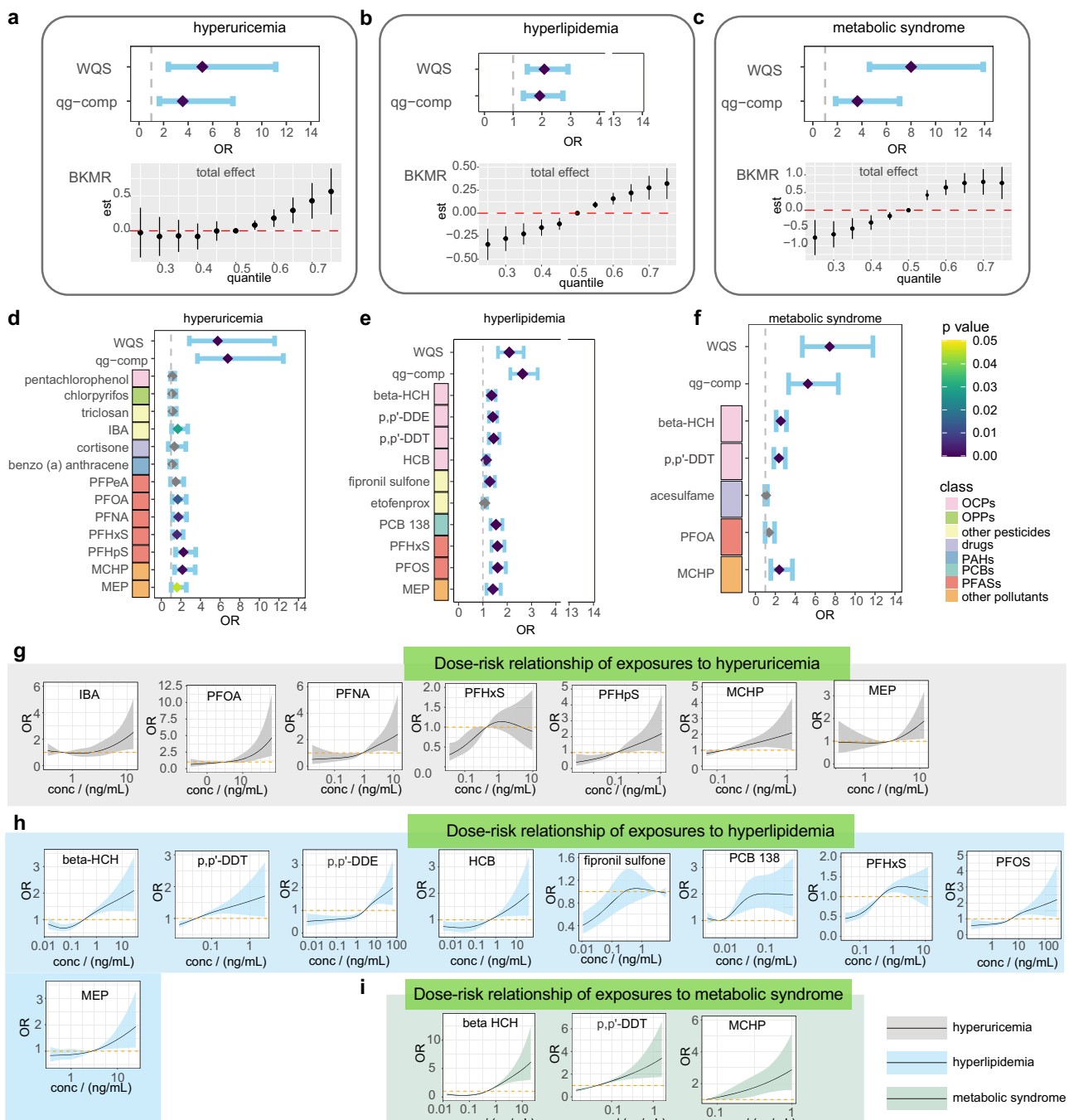

**Fig. 6 | The risk of exposure mixtures and dose-risk relationship between key exposures and related chronic disease.** For three chronic disease outcomes including hyperuricemia (**a**), hyperlipidemia (**b**), and metabolic syndrome (**c**), all exposed mixtures have a positive risk effect on them based on WQS, q g-comp, and BKMR Models. **d**–**f** Odds ratios (ORs) of the joint and each of the priority risk chemicals screened by three multi-exposure models (The chemicals given in figure were defined in at least two models). ORs of the jointed chemicals were obtained by WQS and q g-comp models, and OR of each chemical was obtained by binary logistic regression model. ORs represent odds ratios per one-unit increase in log-transformed exposure mixtures or single exposure levels. All of the three multi-exposure models adjusted for age, gender, region, smoking, and drinking history, and the binary logistic regression model adjusted for age, gender, region, sampling time, education and income levels, marital status, smoking, and drinking history. The position and color of diamond represent ORs and significant (two-sided), respectively. Gray diamonds represent no significance. Error bars represent 95% confidence interval. **g** Dose-risk relationship of exposures to hyperuricemia, specifically, seven key exposures screened by both single and mixed models. **h** Dose-risk relationship of exposures to hyperlipidemia, nine key exposures were included. **i** Dose-risk relationship of exposures to metabolic syndrome, three key exposures were included. The black solid line represents the OR, and gray, blue and dark green shadow represents the 95 % confidence interval of hyperuricemia ($n = 927$ biologically independent samples), hyperlipidemia ($n = 2842$) and metabolic syndrome ($n = 1284$) ORs, respectively.

Biomonitoring II (HBM II) or Biomonitoring Guidance Values (BGV)) of only 8 chemicals were found (see Supplementary Data 11). The health risks associated with the population exposure levels in our study were evaluated based on hazard quotients (HQs) (Supplementary Fig. 19).

Most exposure levels of PFASs were within the safe range compared to those of RfD, except for PFOA, PFOS, PFNA, and PFUnDA, which approached or exceeded risk concentrations in a few individuals (Supplementary Fig. 19a). Individuals had excessive exposure

reference values (BE, HBM II and BGV) for HCB, PFOA, and PFOS, suggesting potential health risks (Supplementary Fig. 19b). For hyperlipidemia risk, dose–risk relationship curves revealed 16 related chemical exposure guidance values. The PFOA and PFOS reference concentrations are 3.19 and 5.26 ng/mL, respectively, lower than the European reference concentrations of 10 ng/mL and 20 ng/mL, respectively, for HBM II. In addition, 5 chemical exposure reference values were not reported, including beta-HCH, p,p'-DDE, PFHpS, 6:2 Cl-PFAES, and MEP (Supplementary Fig. 19c).

### Risk effects of exposure mixtures on related chronic diseases

Exposure to chemical mixtures is a real-life scenario of human exposure but has received insufficient attention in recent studies. In our study, associations of exposure mixtures with each disease outcome were explored by three multi-exposure models, including weighted quantile sum regression (WQS)[34], quantile g calculation (q g-comp)[35], and Bayesian kernel machine regression (BKMR)[36,37]. The results showed that high-frequency exposure mixtures had adverse effects on hyperuricemia, hyperlipidemia and metabolic syndrome (Fig. 6; Supplementary Data 12). The risk chemicals that contributed more to the model were obtained by weights and posterior inclusion probability (PIP). For hyperuricemia, 14, 10 and 13 preferred risk chemicals were obtained by the above three models (Supplementary Fig. 20a–d), respectively, with 13 chemicals overlapping in at least two models (Supplementary Fig. 20e). For hyperlipidemia, 15, 7, and 10 important risk chemicals were obtained by the three models, respectively (Supplementary Fig. 20f–i), with an overlap of 10 chemicals (Supplementary Fig. 20j). For metabolic syndrome, 11, 7, and 6 significant risk chemicals were obtained, respectively (Supplementary Fig. 20k–n), with an overlap of 5 chemicals (Supplementary Fig. 20o). The overall risks of high-frequency exposure mixtures obtained from the three models are shown in Fig. 6a–c.

The chemicals overlapping in the above 3 multi-exposure models were considered as priority risk chemicals and included mainly OCPs (beta-HCH, p,p'-DDT, p,p'-DDE, HCB, pentachlorophenol), PFASs (PFPeA, PFOA, PFDA, PFHxS, PFHpS, PFOS), phthalates (MCHP, monocyclohexyl phthalate (MEP)) and other pesticides (IBA, fipronil sulfone, chlorpyrifos, triclosan, etofenprox) (Fig. 6d–i; Supplementary Fig. 21). Mixtures of these priority risk chemicals were reincorporated into the WQS and q g-comp models to obtain ORs, which were found to be significantly greater for mixture exposure than for single chemicals. This indicated that the mixtures had a nonnegligible risk-enhancing effect. There are several priority risk chemicals with no significant risk according to the single-exposure model (Fig. 6d–f; Supplementary Fig. 21). This suggests that the multi-exposure model is more sensitive in identifying risk chemicals than the other models and can be used as a complement to single-exposure risk analysis.

To better understand the dose–risk relationship, overlapping priority risk chemicals were analyzed, and they presented monotonically increasing nonlinear associations with the risk of chronic disease (Fig. 6g–i). Specifically, 7 of the 13 overlapping priority risk chemicals for hyperuricemia had significant risk effects according to the single model: IBA, MCHP, MEP, PFOA, PFNA, PFHxS and PFHpS. All exposures showed a rapid increase in risk with increasing concentration, except for PFHxS, which showed a faster increase in risk only at low concentrations (Fig. 6g). Nine of the 10 overlapping priority risk chemicals for hyperlipidemia had significant risk effects according to a single model: beta-HCH, p,p'-DDT, p,p'-DDE, HCB, fipronil sulfone, PCB 138, MEP, PFOS, and PFHxS. Among them, fipronil sulfone, PCB 138 and PFHxS showed a faster increase in risk only at low concentrations, and the others increased more rapidly at higher concentrations (Fig. 6h). Three of the 5 overlapping priority risk chemicals for metabolic syndrome had significant risk effects according to a single model: beta-HCH, p,p'-DDT, and MCHP. All exhibited a faster increase in risk at high concentrations (Fig. 6i). The risk relationships of chemicals identified

only by multi-exposure models were almost nonmonotonic (Supplementary Fig. 21), which may be the reason why they cannot be identified by traditional single-exposure models, such as the linear and logistic regression models used in this study.

## Discussion

In this work, we studied the concentration, distribution and disease risk associated with serum exposure in 5696 control individuals and patients via human biomonitoring. Epidemiological factors such as region, age, and sex significantly influence human exposure levels. For exposure risk, hyperlipidemia, metabolic syndrome and hyperuricemia were associated with multiple exposures. In addition, exposure to OCPs, PFASs, PCBs, and phthalates showed nonspecific associations with risk and nonlinear dose-dependent relationships with chronic diseases. To our knowledge, this study is the largest study to date to examine the association between human serum exposure and health outcomes in terms of both chemical coverage and population sample size (Supplementary Data 13). This study not only provides the most representative biomonitoring data for the issue of exposure in the Chinese population but also provides a broad understanding of the chemicals present in humans and their risk of chronic diseases.

Region was the most influential factor on exposure levels, especially for OCPs, PFASs, PCBs, and PAHs, which exhibited significant differences in population levels across 15 provinces in China (Fig. 3a, b; Supplementary Fig. 9). Among them, PFASs, PCBs, and PAHs are mostly produced through industrial use, including anti-staining materials, electronic waste, and incomplete fuel combustion; OCPs are typical pesticides. Above all, the residue levels of these exposures in humans are closely related to industrial and agricultural development, and exposure levels are greater in industrially developed coastal areas than in inland areas. This finding is consistent with previous findings[38]. This may be the result of the rapid industrialization and urbanization in coastal areas[39]. Notably, the highest accumulation levels of almost all chemical categories were found in the population in the Yangtze River Delta. The most likely explanation is that persistent organic pollutants (POPs) cannot be effectively removed from wastewater[40] and are transferred to downstream areas through water flows[41]. In addition, the potential health risk through the food chain is of concern, as studies have shown that the concentrations of PCBs in most animal-derived food groups in coastal areas are significantly greater than those in inland areas[39]. Therefore, management of pollution sources should be strengthened; otherwise, serious contamination in downstream areas and further health hazards may occur through food chain accumulation in the human body. The results from the cohort in this study provide the most representative data for chemical biomonitoring in the Chinese population to date. The cohort in this study included people from 15 provinces, including southern and northern regions as well as coastal and inland areas across the country, representing 56% of the Chinese population. The results showed that the level of OCPs in human blood is significantly greater in China than in other countries, which is closely related to the national conditions of agricultural development in China. Most PFASs show the highest serum concentrations in China and Korea, with a few species of PFASs having the highest concentrations in human serum in the United States and Canada. This is closely related to the geographical shift of industrial sources from North America and Europe to emerging Asian economies, especially China, since 2002[42].

Age was the second most important factor affecting serum chemical levels. The results showed that age and multiple serum chemical concentrations correlated positively, which was consistent with the findings of others, specifically for OCPs[6,7,15,16,43], PCBs[7,15,16], and PFASs[14,17,18]. The age-dependent association of these POPs may be caused by their excretion rates and exposure histories in different age ranges of the population[6]. Higher levels of exposure to multiple chemicals were found in populations with higher education levels and

incomes, which is consistent with the findings of other studies in specific categories, including PCBs[15], OCPs[6], PFASs[19,44] and phthalates[19]. These findings suggest that social factors may have an impact on exposure to POPs[6]. Moreover, sex was found to be an important factor influencing accumulation and excretion of exposures, and residue levels of more fat-soluble OCPs were greater in women than in men, which may be attributed to the generally high body fat percentage in women[6,45]. On the other hand, concentrations of nonfat-soluble chemicals such as phthalates and PFASs were significantly greater in men than in women, which is largely attributed to the unique menstrual excretions of females[46]. Finally, higher serum levels of PFASs were detected in alcohol drinkers, which has not been reported in previous studies; future studies should continuously focus on the relationship between drinking and multiple environmental chemicals.

In the exposure and health effects study, associations of hyperlipidemia, metabolic syndrome and hyperuricemia with various exposures were found (Figs. 4–6). For hyperlipidemia, the present study showed associations with OCPs, PCBs, PFASs, and phthalates. Our findings validate several previous studies; for example, increased concentrations of PFOA, PFNA[31], PFHxS[47], OCPs and PCBs are associated with elevated total serum lipids, total cholesterol and triglycerides[29,48]. Nevertheless, some new risk effects of exposure were observed in this study. For example, among PFASs, new risk associations were found, including PFHpS, PFOS and 6:2 chlorinated polyfluoroalkyl ether sulfonate for hyperlipidemia. In particular, 6:2 chlorinated polyfluoroalkyl ether sulfonate is a new type of PFAS with few related disease risk association studies. Moreover, few existing studies have focused on the finding that fipronil sulfone significantly increases the risk of hyperlipidemia, possibly because of insufficient biomonitoring coverage, and our study extends previous findings.

Metabolic syndrome is another chronic disease strongly associated with exposure and is a pathological state involving multiple metabolic diseases, including abdominal obesity, dyslipidemia, hypertension, and diabetes[49]. Previous studies have shown an increased risk of metabolic syndrome with various POPs, such as OCPs, PCBs[50], and PFAs[31], which was consistent with our findings. In addition, MEP was shown to be associated with an increased risk of hypertriglyceridemia, a disease subtype of metabolic syndrome, in a previous study of exposure to endocrine-disrupting chemicals affecting the risk of metabolic syndrome in adults[51]. An association between MCHP and the risk of metabolic syndrome was also found in our study. Finally, differences in disease susceptibility among different populations were observed in this study but rarely studied in previous researches. There was a stronger effect of exposure on metabolic syndrome in the elderly and in men (Fig. 5b, c). This may be due to differences in excretion and metabolism among different populations. Previous studies have reported that pesticides are excreted more slowly in elderly individuals and in men[6,46]. Moreover, sex-specific associations between exposure and disease-associated lipid changes may explain sex-related differences at the metabolic level[22].

Hyperuricemia, another chronic disease strongly associated with exposure in this study, is a causative agent for a variety of diseases, including gout, kidney stones, and cardiovascular disease[52,53]. The serum uric acid concentration is strongly influenced by environmental factors, and associations between PFASs and hyperuricemia have been reported[32,33,54]. These findings were again verified in the present study, and a new association of PFHpS with hyperuricemia was found. The effects of other categories of environmental chemicals on uric acid are unknown; significant associations between IBA, MEP, and MCHP and hyperuricemia were found based on both single and mixed exposure models (Fig. 6d, g), which have not been reported in other studies.

Multi-exposure models were jointly used to collectively identify combinations of exposures that are associated with significant risk effects for multiple chronic disease outcomes. Some key risk chemicals screened by multi-exposure models were not identified in the single-exposure model (Fig. 6d–f), reflecting the complementary role of the multi-exposure model to the single-exposure model[55]. In addition, three chronic diseases that are most affected by overall exposure and corresponding exposure mixtures, were identified. Three groups of exposure mixtures showed significant risk-enhancing effects on hyperuricemia, hyperlipidemia, and metabolic syndrome, which have not been reported in previous studies. Future studies should pay attention to these chronic diseases and associated exposure mixtures.

Taken together, our results provide comprehensive insights into the residue levels and exposure characteristics of environmental chemicals in human serum, identifying human serum exposures and their specific combinations that are associated with major chronic disease outcomes. These findings provide guidance for further pollution management and protection of susceptible populations. We recognize that our results do not indicate the causal effect of any chemical on adverse outcomes, which requires further investigation. Nevertheless, this study not only demonstrates the potential of the exposome for disease prediction but also provides a useful resource for more in-depth toxicological and longitudinal epidemiological studies.

## Methods

### Study population and epidemiological information

A cross-sectional study comprising 5696 subjects, including 2141 healthy persons and 3555 patients, was conducted in 15 provinces of China. The study was approved by the National Institute for Nutrition and Health, Chinese Center for Disease Control and Prevention (reference No. 201524); the ethics committee was the Chinese Center for Disease Control and Prevention Institutional Review Board. Moreover, written informed consent was obtained from each participant before the study began. A total of 12 chronic disease outcomes were analyzed, ranging from 243 to 1813; these included diabetes, hyperuricemia, obesity, hypercholesterolemia, hypertriglyceridemia, metabolic syndrome, high diastolic blood pressure, high systolic blood pressure, abdominal obesity, hypertension, high low-density lipoprotein cholesterol and hyperlipidemia (Supplementary Fig. 12b). The study population in this work is a subset of the cohort recruited from the China Nutrition and Health Survey in 2015. The samples covered different regions, sexes, age groups, and various chronic diseases. The details of the China Nutrition and Health Survey were described in previous studies[56,57]. In brief, the cohort was selected using a multi-stage random cluster design and included 360 communities from 15 provinces with varying income levels. Then, 20 families were randomly selected from each community to participate in the study. Well-trained researchers collected epidemiological data and blood samples, and clinical parameters related to chronic diseases were determined through biochemical assays of the blood samples.

Epidemiological factor information, including sex, age, sampling location, sampling time, education level, income level, marital status, cigarette smoking status, alcohol consumption status and medical history, was collected via questionnaire. Environmental factor information, including air pollution and meteorological conditions, was obtained based on regional and sampling time (month) information from open-source websites. Moreover, 9 chronic disease-related clinical parameters, namely, uric acid levels, glycated hemoglobin levels, low-density lipoprotein cholesterol levels, triglyceride levels, total cholesterol levels, systolic blood pressure, diastolic blood pressure, waist circumference and BMI, were determined. The detailed information is provided in Table 1. Twelve chronic disease outcomes were subdivided based on these nine related clinical parameters. A total of ten rounds of survey were conducted on the cohort from 1989 to 2015. For the 2015 survey, 14,000 samples were collected, and we randomly selected half of these samples for exposome analysis. After excluding samples with insufficient volume or incomplete information, a total of 5696 samples were ultimately analyzed in this study.

## The specific criteria for classification of 12 chronic disease outcomes by 9 clinical parameters

1. Individuals with uric acid levels equal to or greater than 420 μmol/L for males, and equal to or greater than 360 μmol/L for females are determined to have hyperuricemia.
2. Individuals with glycated hemoglobin levels equal to or greater than 6.5% are determined to have diabetes.
3. Individuals with low-density lipoprotein cholesterol levels equal to or greater than 3.4 mmol/L are determined to have hyperLDL-C.
4. Individuals with triglyceride concentrations equal to or greater than 2.3 mmol/L are determined to have hypertriglyceridemia.
5. Individuals with total cholesterol levels equal to or greater than 6.2 mmol/L are determined to have hypercholesterolemia.
6. Individuals with systolic blood pressure equal to or greater than 140 mmHg are determined to have high systolic blood pressure.
7. Individuals with diastolic blood pressure equal to or greater than 90 mmHg are determined to have high diastolic blood pressure.
8. Individuals with waist circumference equal to or greater than 90 cm for males, and equal to or greater than 85 cm for females are determined to have abdominal obesity.
9. Individuals with a BMI (Body Mass Index) greater than 28 kg/m2 are determined to be obese.
10. Hypertension is determined if either the systolic blood pressure is equal to or greater than 140 mmHg, or the diastolic blood pressure is equal to or greater than 90 mmHg (refer to Supplementary Fig. 12e).
11. Hyperlipidemia is determined if any of the three clinical indicators are met: low-density lipoprotein cholesterol equal to or greater than 3.4 mmol/L, triglyceride concentration equal to or greater than 2.3 mmol/L, or total cholesterol equal to or greater than 6.2 mmol/L (refer to Supplementary Fig. 12c).
12. Metabolic syndrome is determined if an individual has three or more of the following four chronic diseases: abdominal obesity, hypertriglyceridemia, hypertension, and diabetes (refer to Supplementary Fig. 12d).

### Environmental variable estimates

Environmental factors including air pollution and meteorological conditions were obtained using regional and sampling time (month) information from open-source websites. Specifically, for air pollution, data on three variables including Air Quality Index (AQI), PM2.5, and PM10 were collected from a recent study[58] with open data on website https://quotsoft.net/air/. Daily data for each region (specific to city/county) were available, and based on these data, monthly values of AQI, PM2.5, and PM10 were calculated for each individual in their respective region (specific to city/county) and sampling month.

For meteorological conditions, data on three temperature variables including daily maximum temperature, daily minimum temperature, and daily mean temperature were collected from a surface climate dataset of China in the website[59] https://www.geodoi.ac.cn/WebCn/doi.aspx?Id=3187. Daily data for each region (specific to province) were accessible, and based on these data, monthly mean values of Daily_Maximum_Temp, Daily_Mean_Temp, and Minimum_Temp were calculated for each individual in their respective region (specific to province) and sampling month.

Finally, the obtained six environmental variables and nine epidemiological factors were treated as confounding factors and adjusted by binary logistic regression model.

### Chemicals and reagents

Ultrapure water was prepared with a Milli-Q water purification system (Millipore, 7 Billerica, MA, USA). HPLC grade acetonitrile and methanol were obtained from Merck (Darmstadt, Germany). Ammonium acetate and fetal bovine serum were purchased from Thermo Fisher Scientific (Rockford, USA). Formic acid was purchased from National Medicines Corporation Ltd. (Beijing, China). Dichloromethane was purchased from J.T. Baker (Rockford, USA). Hexane was purchased from Merck Sigma–Aldrich (Darmstadt, Germany). The 267 chemical standards for each analyte were acquired from Alta Scientific Co., Ltd. (Tianjin, China) (Supplementary Data 1). Twenty-seven internal standards were isotope-labeled chemical standards, which were purchased from several companies; the detailed information is provided in Supplementary Data 14.

### Selection of 267 target chemicals

The principles of prioritized list selection and identification were described in our previous study[60]. Briefly, at least one of the following three conditions should be included. First, chemicals, including pesticides, veterinary drugs and POPs, are often reported in the literatures with high concentration levels and high detection frequency in blood (plasma/serum). Second, according to the Integrated Risk Information System and the International Agency for Research on Cancer, these exposures can have health effects or carcinogenicity. Third, considering their dietary exposure risks, chemicals are often found to exceed residue limits in daily food in routine assays by the authority agency. Based on the above principles, the detection scope was expanded in our study, and two platforms (GC–MS/MS and LC–MS/MS) were used to cover prior exposures comprehensively. After grouping by region, chemicals with a detection frequency greater than 50% in any one of 15 provinces was defined as having high-frequency exposure and were the focus of subsequent analysis.

### Method development and analysis based on GC–MS/MS

The method used for the GC–MS/MS assay reported in a previous study[61] was modified, and the target analytes were expanded from 35 to 97. Before GC–MS/MS analysis, solid-phase extraction (SPE) was used for pretreatment of serum samples. The specific pretreatment steps were as follows: (a) Adding internal standard: 10 μL isotope standard (Supplementary Data 14) mixture was added to 200 μL serum samples and stored at 4 °C overnight for later use. (b) Deproteinization: 200 μL of 15% formic acid aqueous solution was added to the above serum sample, after which the sample was vortexed. (c) Activation of SPE cartridges: before adding the serum sample, 3 mL of dichloromethane, 3 mL of methanol, and 3 mL of ultrapure water were added to Oasis® PRiME HLB cartridges in advance. (d) Adding samples: all the above pre-treated serum samples were transferred to the Oasis® PRiME HLB cartridges. (e) Rinsing and vacuuming: the sample tube was rinsed twice with 1 mL of methanol:water (1:6, v/v) at a flow rate of 0.5–1 mL/min, and the SPE cartridges were rinsed with 1 mL of methanol:water (1:6, v/v) and evacuated for 20 min. (f) Elution: the samples were eluted with 3 mL of dichloromethane and 3 mL of n-hexane, with all the eluent collected. (g) Nitrogen blowing and redissolving: the eluent was nitrogen-blown nearly dry, redissolved in 100 μL of acetone, and transferred into a sample bottle for later analysis.

### Parameter settings of target method based on GC-MS/MS

The experiments were performed using an 8890 GC system equipped with a DB-5MS column (30 m × 0.25 mm × 0.25 μm; Agilent, Santa Clara, USA) coupled to a Agilent 7000D triple quadrupole mass spectrometer. For GC system, high purity helium (99.999%) was used as carrier gas with the flow rate of 1.2 mL/min. The injection volume was 1 μL, and was conducted in splitless mode at 270 °C. The analytes were separated by temperature programming: the initial temperature increased from 70 °C to 150 °C at the speed of 25 °C/min, then increased to 200 °C at 3 °C/min and kept for 2 min. Finally, the temperature was increased to 300 °C at 8 °C/min and kept for 6 min. For MS/MS condition: the MS was equipped with electron bombardment

ion source with the ionization voltage of 70 eV. The temperature of ion source was 300°C, the temperature of transmission line was 300 °C, and the detector gain was 0.90 kV + 0.30 kV. Quantitative analysis was performed by using multi-reaction monitoring mode with the solvent delay of 6 min. The detailed mass spectrum parameters are presented in Supplementary Data 15.

## Method development and analysis based on LC–MS/MS

The serum sample pretreatment method was modified from a previously reported method[60], and the target analytes were expanded from 106 to 170. In brief, a high-throughput sample processing method was used to extract samples from serum. Deproteinization and purification of serum were conducted using a phospholipid removal plate (Phenomenex, Torrance, USA) with 96 wells. A total of 280 μL of acetonitrile containing 19 internal standards (Supplementary Data 14) was added to each well and mixed with 70 μL of the serum sample. The 96-well filter plates were covered with aluminum foil and shaken for 10 min at room temperature. Proteins and phospholipids were removed after centrifugation at 1000 × g for 10 min at 4 °C. The supernatant was concentrated with nitrogen flow and reconstituted with 70 μL of methanol/water (1:1, v/v) as the solvent. The final extract was filtered through 0.22 μm centrifugal filters (Biotage, Uppsala, Sweden). Finally, 5 μL of filter liquor was subjected to LC–MS/MS.

## Parameter settings of target method based on LC-MS/MS

Targeted analysis was performed on a Exion LC AD ultrahigh-performance liquid chromatography (UHPLC) (AB SCIEX, Framingham, U.S.A) coupled with triple-quadrupole 6500 plus mass spectrometry (AB SCIEX, Framingham, U.S.A). The separation was conducted on an ACQUITY UPLC BEH C18 Column (Waters, Milford, MA) 2.1 × 50 mm 1.7 μm at 60 °C oven temperature. An ACQUITY UPLC® BEH C18 VanGuardTM Pre-Column (Waters, Milford, MA) 2.1 × 5 mm 1.7 μm was added before analytical column. The flow rate was 0.4 mL/min and the injection volume was 5 μL. Mobile phase A was 5 mM Ammonium acetate in water while mobile phase B was 5 mM Ammonium acetate in methanol. The gradient program for mobile phase B was started at 5% C maintained for 0.5 min, linearly increased to 50% in 4.5 min, then linearly increased to 70% for 4 min, linearly increased to 100% for 3 min, held for 2 min, then dropped sharply to 5% for 0.1 min, and held for 2 min. The total run time was 16.0 min. Valve switching was set so that the eluate of samples before 0.5 min did not enter the mass spectrometry. In the mass spectrometer system, ionization of targeted analytes was performed by electrospray ionization with positive/negative switching mode. The electrospray voltage is set at 5500 V for positive ion scanning mode and -4500 V for negative ion scanning mode. The curtain gas is set at 40.0 psi, and the ion source temperature is set at 350 °C. The ion source gases 1 (GS1) and 2 (GS2) are both set at 50 psi. Declustering potential and collision energy voltages were optimized for each chemical based on corresponding standard. The detailed parameter settings are presented in Supplementary Data 16.

## Quality control and assurance

To ensure the stability of the entire analytical process, we applied the following standard operating procedure: (1) The same internal standards were added to each real sample and QC sample. (2) The spiked serum samples were used as QC samples and were inserted after every 21 and 11 real samples for GC–MS/MS and LC–MS/MS, respectively. (3) Routine instrument maintenance, such as blank flushing, linear replacement for the GC–MS/MS platform, needle replacement for the LC–MS/MS platform, cleaning of the ion source, and full maintenance when necessary, was performed before each batch of sample analysis. (4) Multiple calibration curves were used throughout the whole sample analysis process, and each independent calibration curve was run for each batch of samples. For the GC platform, each batch consisted of 158 real samples and 21 QC samples; for the LC platform, each batch

contained 264 samples and 24 QC samples. Hence, there were 29 and 20 independent calibration curves for the GC–MS/MS and LC–MS/MS platforms, respectively, bracketing the entire sample analysis process.

Data quality was evaluated using the following steps. (1) Quantification of the QC samples of the batch using the calibration curve of each batch. (2) Definition of the appropriate internal standard. For the GC platform, each internal standard was used to correct a dozen chemicals within their neighboring retention time regions; for the LC platform, internal standards were selected with the smallest RSD in QCs after internal standard correction[62]. In other words, the defined internal standard was used to correct the corresponding chemical in the calibration curves and QC samples, and the corrected calibration curve was used to quantify the chemicals in the QC samples of the corresponding batch. Finally, the optimal internal standard was selected based on the smallest RSD of the chemicals in the QC samples after correction and quantification. (3) Evaluation of the batch effect with QC samples. For all the QC samples, the mean concentrations of all the chemicals and the principal component score plots were used to evaluate batch effects. The effectiveness of batch effect correction can be determined by comparing the original signals with the batch-specific quantification concentrations and the batch-specific quantification combined with multiple internal standard corrections. A principal component analysis plot was generated using SIMCA-P 14.1 software (Umetrics, Umea, Sweden). (4) Evaluation of the reliability of the quantitative results with respect to the QC samples. Each chemical in the QC was calibrated with the selected internal standard, and the QC and samples of the corresponding batches were quantified by using the calibrated linearity. RSD < 30% for each substance in the QCs was used as the criterion for stable detection. Recovery within 80–120% for each substance in the QCs and calibration curves at low, medium, and high spiked concentrations were used as the criterion for accurate detection.

## Methods examination and evaluation

In order to ensure the stability and feasibility of the established analytical method, methodological evaluation was carried out before conducting large-scale sample testing. The methodological examination and evaluation were reported in previous studies for GC-MS/MS[61] and LC-MS/MS[60] platforms, the present study extended the monitoring list and reanalyzed part of the methodological examination as follows. To determine the limit of detection and linearity of calibration, matrix-matched standard solutions at 0, 0.02, 0.05, 0.1, 0.2, 0.5, 1, 2, 5, 10, 20, 50, and 100 ng/mL were analyzed for GC-MS/MS. As for LC-MS/MS platform, matrix-matched standard solutions at 0, 0.001, 0.0025, 0.005, 0.01, 0.025, 0.05, 0.1, 0.25, 0.5, 1, 2.5, 5, 10, 25, 50, 100, and 200 ng/mL were analyzed. The minimum concentration satisfying a signal-to-noise ratio greater than 10 on the chromatogram was defined as the limit of quantification (LOQ). The linearity of calibration was evaluated (the number of calibrator levels greater than or equal to 5) by the correlation coefficient (r2) of the calibration curve from LOQ to a suitable concentration. A weighting regression factor of 1/x (x represent concentration) was adopted to minimize calculation error at low concentrations[63]. The precision of the two platforms was evaluated using spiked QC samples with the concentrations of 2 ng/mL and 10 ng/mL, respectively. Due to 49 calibration curves bracketing the entire sample analysis process, accuracy tests were performed for each one. For GC-MS/MS platform, the accuracy of each exposure in each calibration curve was calculated at 2, 5, and 20 ng/mL concentration levels, and for LC-MS/MS platform, the accuracy of each exposure in each calibration curve was calculated at 1, 10, and 100 ng/mL concentration levels. The raw method files of GC-MS/MS and LC-MS/MS were presented in Supplementary Software, which can be directly opened and used by Agilent "MassHunter Workstation" and AB SCIEX "Analyst" software, respectively.

## Statistical analysis

For quantitative analysis of chemicals, peak areas of the serum chemicals and internal standards monitored by both the GC–MS/MS and LC–MS/MS platforms were obtained from the raw data by "masshunter" (Agilent, Santa Clara, USA) and "SCIEX OS" (AB SCIEX, Framingham, USA) software, respectively. Both software programs generate quantification curves using a weighted linear regression model ($1/x$) to ensure the accuracy of quantification at low concentrations[63]. The method of internal standard selection and calibration is described in the "QC and assurance" section. The internal standard calibration curve method was used for quantification. Concentrations less than the LOQ were set as LOQ/√2.

Quantification process of the internal standard calibration curve method: The internal standard calibration curve method was applied for quantification, which is commonly used in precise quantitative research of endogenous metabolites and exogenous chemicals in biological matrices. The point of this method is a group of consistent internal standard (isotopic labeled) mixture was added to both the calibration curve and each sample, and quantification was achieved through regression equations between peak area ratio and concentration. The specific process is as follows:

1. Define the appropriate internal standard. For the GC platform, internal standard was selected according to similarities in chemical properties and retention behavior, and used to correct chemicals within its neighboring retention time regions. For the LC platform, the internal standards were selected according to the rule of smallest RSD in QCs after internal standard correction (details see "Quality control and assurance" section). 2. Construct calibration curves after internal standard (IS) correction. Concentrations of analytes in calibration curve (Cc) were set as the independent variable, the ratios between peak areas and corresponding internal standard are set as the dependent variable (AREA Cc / AREA IS). A weighted linear regression model (weight = $1/x$) was used to construct calibration curves. 3. Obtain relative responses of the samples after IS calibration. Relative responses (AREA sample / AREA IS) were the ratio between the peak area of the sample (AREA sample) and the peak area of the corresponding internal standard (AREA IS). 4. Calculate the concentration of the analyte in samples. Relative responses of samples (step 3) are substituted into the calibration curve in step 2 to obtain the sample concentration.

In addition, to ensure quantitative accuracy, a calibration curve was inserted at the beginning of each batch as mentioned before, therefore, the quantitative methods described above were further performed in batch-specific way, means that "each calibration curve was used to quantify the actual samples in its own batch." This practice helps eliminate errors introduced by pauses in routine instrument maintenance. For the few analytes not assigned to the internal standard with smallest RSD in QCs, their quantification was achieved by external standard method.

Effect of epidemiological factors on the exposome: To assess which epidemiological factor influenced the overall exposure variation, variation partitioning analysis was performed using the R package "vegan", and the exposure levels used were log10 transformed. Principal component analysis was carried out with the R package "FactoMineR". To investigate the correlation between epidemiological factors and each chemical, PASW Statistics 18 software (SPSS, Chicago, IL) was used to conduct partial Spearman correlation analysis between one epidemiological factor and each chemical; the other eight epidemiological factors were used as confounders. The correlation coefficients were used to construct a correlation heatmap matrix and a correlation network diagram (Cytoscape software 3.7.1).

Stratified analysis of exposure characteristics: To explore differences in serum residue levels in different populations, samples were stratified based on each epidemiological factor. Exposures that differed significantly among groups are shown in Fig. 3. First, for continuous and ordinal categorical variables, including age, education, and income, significantly correlated chemicals were obtained using partial Spearman correlation and multiple linear regression while controlling for the false discovery rate for multiple corrections ($p$ & FDR < 0.05). To exclude the interference of age on education level, data for those older 30 years of age were selected for analysis. Significantly different exposures are shown in a line graph using scaled concentration. Second, for binary epidemiological factors such as sex, smoking status, and alcohol consumption, the $p$ values of nonparametric tests were adjusted by controlling for the false discovery rate ($p$ and FDR < 0.05). Considering the lack of data on female smoking and drinking habits, relationships between smoking and drinking and residue concentration were analyzed only for males. Significantly different exposures are shown in bar plots using the fold change in the geometric means. All statistical tests were two-sided. Third, geometric means were used to reduce the effect of extreme values when studying the regional distribution of exposures, and geometric means were obtained by PASW Statistics 18 software (SPSS, Chicago, IL). A regional heatmap and stacked bar plot were generated to visualize the regional distribution of exposures. A regional heatmap was generated with https://www.bioinformatics.com.cn, an online platform for data analysis and visualization. Other figures were generated with the package "ggplot2" in R software (version 4.2.1, R Foundation for Statistical Computing, Austria).

Risk of disease for each individual chemical: To adjust for confounders, the R package "MatchIt" was used to match a control sample for each disease, and patients with more than 2 diseases were not considered, except for metabolic syndrome; a control group was matched based on all nine epidemiological factors and five major chronic diseases (obesity, hypertension, diabetes, hyperuricemia, and hyperlipidemia). Subsequently, the risk of each chemical for disease was analyzed using binary logistic regression and multiple linear regression and again adjusted for the nine epidemiological factors as confounders. To understand susceptibility to disease risk from exposure in different age and sex populations, samples were grouped by age and sex. Specifically, the population was divided into three age groups: 30–50 years, 50–60 years, and 60–80 years. The participants were also divided into male and female groups. Subsequently, propensity score matching was used to classify the data sets into disease and control groups for each subgroup. Finally, binary logistic regression was used to determine ORs for the exposures in each group. The exposure concentrations used were log10 transformed for all regression analyses. Two-sided $t$ tests and Hosmer–Lemeshow tests were employed for multiple linear regression and binary logistic regression, respectively.

Health risk assessment: First, available exposure guidance values were collected including BE, HBM II, BGV, and RfD[64]. Then health risk assessments were carried out using the HQ[65]: HQ = $C_{serum}$/$C_{guidance}$. HQ > 1 suggested exposure levels exceeding published human health benchmarks. Dose-response curve on the one hand can display the linear or nonlinear relationship between exposure dose and risk; on the other hand, a curve can identify the minimum exposure dose associated with increased disease risk (OR > 1). The R package "RSC" was used to determine the dose–risk relationship of key exposures in this study.

Exposure mixture to disease risk: We used a combination of WQS[34], q g-comp[35], and BKMR[36,37] to assess the association of exposure mixtures with multiple chronic diseases. Each model was adjusted for the covariates region, age, sex, smoking status, and alcohol use. Unlike the univariate analysis described above, which used 4756 samples from the GC platform and 5513 samples from the LC platform, 4573 common samples were selected for mixed exposures, and propensity score matching was applied to identify case controls. In addition, 35 individuals with a detection frequency > 50% nationwide were selected for analysis to meet the model quartiles. The exposure levels used were log10 transformed and then scaled for multi-exposure analysis.

## Brief description of three multi-exposure models

The weighted quantile sum regression (WQS) scores were estimated using the R package "gWQS"[34], which groups different chemicals into ordinal variables (quartiles), and calculated a weighted linear index through the WQS regression model, which represents the entire body burden of all chemicals. the WQS regression estimates sum mixture effects in either positive or negative directions, respectively, so the likelihood of an association was assessed in both directions in separate models. 1000 bootstrap runs were performed for each analysis to optimize the association between WQS scores and outcomes in the multivariate linear regression model. Ultimately, the model provides an estimate beta and significance of the total effect and the corresponding weight for each chemical, which shows how much a particular chemical contributes to the WQS index. Odds ratios (ORs) were calculated according to formula: $OR = exp$ (beta). 95% confidence intervals of ORs were calculated according to formula: upper bound of $OR = OR + standard\ error\ (OR) \times 1.96$, lower bound of $OR = OR - standard\ error\ (OR) \times 1.96$.

Mixture effects was estimated using the R package "qgcomp"[35], which is similar to the WQS in that different chemicals are grouped into ordinal variables (quartiles) and a weighted linear index representing the cumulative effect is estimated by the regression model. The model also provides estimates and significance of the total effect and the corresponding weights for each chemical. The difference is that the WQS regression estimates are performed in either positive or negative directions separately whereas quantile g-computation (q g-comp) allows the joint effects of different directions of individual exposure to be assessed simultaneously in a single run. Moreover, the run speed can be greatly improved by G-computation and more robust associations than the WQS model can be obtained in small sample sizes.

BKMR, a non-parametric Bayesian variable selection framework, can investigate flexibly the joint effects of exposure mixtures on human health[36,37]. BKMR model can provides overall risk, single variable risk and non-linearity interaction of exposure to responses, and PIP for each exposure. Among them, PIP describe the relative importance of each exposure to the outcome of interest. Here, hierarchical variable selection method was used due to highly correlated exposures, then groupPIP and conditional PIPs (condPIP) were obtained to commonly evaluate the contributor of exposures. BKMR model was conducted with R package 'bkmr' using the binomial link function for binary outcomes and 5000 iterations using a Markov chain Monte Carlo algorithm to ensure convergence.

## Reporting summary

Further information on research design is available in the Nature Portfolio Reporting Summary linked to this article.

## Data availability

All data supporting the findings of this study are available within the paper, in the supplementary information file, and in the source data file. The air pollution dataset of China can be found in https://quotsoft.net/air/, and the surface climate dataset of China can be found in https://www.geodoi.ac.cn/WebCn/doi.aspx?Id=3187. The concentration levels of 74 high-frequency exposures in human serum of Chinese chronic diseases population have been given in Table 2 of the paper, but the generated individual exposure atlas data are considered sensitive biomonitoring data, therefore, can not be publicly available according to the contracts with cooperating institutions (the initiator of the cohort) and the limitations included in the informed consents signed by the study participants. The request of these individual data is suggested by sending an email to the corresponding author Dr. Guowang Xu (xugw@dicp.ac.cn). Requests should include name, affiliation and contact details of the person requesting the data, which data are requested and the purpose of requesting the data. Requests will be subject to consideration by the management committee of the corresponding institutes and the sample collection institutes, including Dalian Institute of Chemical Physics, Chinese Academy of Sciences, National Institute for Nutrition and Health, Chinese Center for Disease Control and Prevention, Huazhong University of Science and Technology. If approved, the corresponding author will send the request data by email. Time frame for a response will be within 3 months. Data requests under agreement will be considered for purposes of reproducing the data and subject to appropriate confidentiality obligations and restrictions. Applicants must promise that these individual data applied for will only be used for scientific research and cannot be publicly released. Source data are provided with this paper.

## Code availability

No custom code was used in this study. The R codes for statistical analysis and figure production have been deposited to the GitHub and Zenodo (https://doi.org/10.5281/zenodo.10391262[66]).

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

## Acknowledgements

This research was supported by the National Key R&D Program of China (2019YFC1605100, G.X.), the National Natural Science Foundation of China (No. 21934006, G.X.), Youth Innovation Promotion Association CAS (2021186, Xinyu Liu), and the innovation program (DICP ZZBS201804, G.X.) of science and research from the DICP, CAS.

## Author contributions

G.X., Xinyu Liu. and S.M. developed the concept and designed experiments. C.S. collected and provided the human serum samples. L.Y. M.W. and G.J. performed the experiments of LC-MS/MS part and analyzed the data. F.Z. contributed to data analysis. J.K., Xiang Li., and M.Z. performed the experiments of GC-MS/MS part and analyzed the data. Y.W., T.C., T.L., L.Z., X.S., and C.Z. provided significant intellectual input. L.Y. drafted the manuscript. Xinyu Liu., G.X., and S.M. contributes to article revisions.

## Competing interests

The authors declare no competing interests.
