## [Peer Review file · Nature Communications]

REVIEWER COMMENTS

Reviewer #1 (Remarks to the Author):

- What are the noteworthy results?

- o Within this study, the authors characterized a subset of the serum exposome (267 chemicals) for a study cohort spanning across China (n = 5696) via targeted GC-MS/MS and LC-MS/MS analyses. This include quantitation results with concentration levels for chemicals observed. The study cohort also included demographic data and health data, which allowed authors to perform statistical analyses in order to identify potential indicators (i.e. sources) of chemical exposure, as well as identify specific linkages between the presence of chemical and adverse health indicators as well as disease outcomes. This represents a compelling data set with enough size and diversity to identify compelling associations that may ultimately inform where chemical exposures are coming from, and which ones are concerning.

- Will the work be of significance to the field and related fields? How does it compare to the established literature? If the work is not original, please provide relevant references.

- o Exposomics is increasingly of importance as it has been found that many of these chemical exposures encountered are related with the onset of disease. With that said, it is still being established how best to utilize/integrate chemical results with supporting metadata in order to identify when an observed chemical/chemical level is significant. The authors in this study use a nation-wide cohort to characterize part of the exposome in order to have a diversity in study population that can be utilized when perform statistical analyses on chemical results.

- o A limitation of this study is that the "exposome", or the totality of chemical exposures an individual encounters, is broad to the point of being currently primarily undefined, and in efforts of characterizing it most exposomics studies utilize non-targeted analysis/suspect screening with large libraries of chemicals. This study performed targeted analyses with 267 chemicals, which allowed for defined chemical results with concentration levels but also is very limited in scope when determining the "exposome" of a population.

- o Also, the criteria for the selection of the 267 chemicals included previous associations with health risks according to IRIS/IRAC, as such, the scope of this study was not about identifying new chemicals of concern from a public health standpoint, but further characterizing potential associations of chemicals with already known potential hazard. This work still however represents a large pool of data from which the authors have mined some compelling associations, and from which potentially others could benefit from analyzing statistically.

- o This work represents quantitative data for 267 chemicals. Have these concentration levels been compared with tox data available for these chemicals in order to determine if there was an overlap between the levels observed, and the levels that would be significant from a health effect?

- Does the work support the conclusions and claims, or is additional evidence needed?

- o The authors have described work that communicates the observation of 267 chemicals via targeted methods along with statistical associations with potential chemical source paths as well as potential health risks/disease outcomes. There are some questions regarding the acquisition of the targeted data (accuracy, batch effects, internal standards- see below) that must be addressed in order to determine if the results are appropriate for the conclusions drawn.

- Are there any flaws in the data analysis, interpretation and conclusions? Do these prohibit publication or require revision?

- o The authors selected 74 exposure chemicals from the 267 initially analyzed because these were more prevalent across the study population (detected in >50% of samples). While this may down-select for chemicals with greater occurrences and potentially greater statistical significance, this seems unnecessarily conservative and restrictive for determining links between chemicals, exposure sources and health effects. While it is true that a greater presence of exposure across a population could be more significant if there are health risks associated with that exposure, it does not necessarily mean that exposure is of greater individual health risk. There is potential that the 197 chemicals not included with analyses with demographic data and health effects may actually identify some very meaningful exposure sources and/or disease linkages of even greater importance than the more prevalent 74 chemicals. If possible, it would be recommended that the authors include the 197 chemicals in their statistical analyses performed.

- Is the methodology sound? Does the work meet the expected standards in your field?

- o Typically for targeted analyses, calibration standards bracket the analytical run rather than solely being run at the beginning as was done in this study. While QC samples were run throughout the analytical run for verification of instrument stability, this has the potential to skew results for samples acquired in the beginning vs. end of a run, which is particularly concerning if no sample randomization was performed.

- o Additionally, labelled internal standards can be used in targeted analyses to correct for matrix effects and instrument variability throughout the run. In order to most accurately correct for these effects, the labelled internal standards are usually labelled versions of the target analytes but in this study a set of labelled non-target analytes were chosen as internal standards, which the specific IS compounds inducing the lowest variability being selected for use as a given analyte's IS. This has potential to affect results, particularly for multiple batches of analytical runs.

- o Analytical QC's were used to validate each analytical run. Typically these are used to assess accuracy and precision of the run. The methodology describes a criterion of <30% RSD for the QCs but makes no mention of the accuracy of the QC's. Again, this is particularly significant considering the fact that the calibration standards were run only at the beginning of the run; instrument drift throughout a run could potentially result in inaccurate (but precise) results that would pass the criteria used in this study.

- Is there enough detail provided in the methods for the work to be reproduced?

- o It is unclear whether there was any sort of sample randomization performed when performing the MS analyses (see above comment on calibration standard bracketing)

- o It is unclear if any batch effects were observed in the different analytical runs performed, and if so, if any batch correction was applied. This is of significance for such a large study run across multiple sampling times and analytical runs. Particularly considering for example, regions and sampling time were found to have the largest variations in exposures, but if they were run at different times batch effects may have contributed to these variations

General suggestions:

- It would be recommended for the authors to revise some of the text as some of the language used may not accurately represent what is meant to be communicated. Primarily, there are two areas:

- o References to the exposome: the exposome represents the totality of chemical exposures that an individual encounters, which is a very broad scope of chemicals. It has been referenced in the text as being characterized by this study, when in fact it is only characterizing a subset or portion of the exposome as it only monitors 267 chemicals. For example, see lines 125-126 in the text: "Then, the human serum exposome including 267 chemicals was comprehensively determined by GC-MS/MS and LC-MS/MS platforms". Consider reviewing references to the exposome in the text to reflect that this study is only characterizing a portion of the exposome.

- o The authors include disclaimer text at the end of the discussion indicating that this study does not provide causality of chemicals on adverse outcomes. However there are references in the text in which the language reflects causality, i.e. lines 251-253: "Specifically, OCPs and PCBs increase the risk of hypertension, diabetes, metabolic syndrome and obesity. PFASs increase the risk of hyperlipidemia, metabolic syndrome, diabetes, and hyperuricemia". It would be suggested that the author review the text (primarily results and discussion) such that this language is updated such that it does not suggest causality.

- This study represents a very compelling collection of data of which the authors have done statistical analyses to identify risk/health associations, but would also serve others by being available to other researchers. If the analytical data (267 chemicals across 5696 individuals) along with demographic data were provided in supplemental, while this would be a large table it would provide value to other researchers.

Lines 190-191: Please provide literature reference for reported values of PFAS in Yangtze.

Lines 200-202: Please provide references/sources of these literature values.

Reviewer #2 (Remarks to the Author):

This is a really interesting paper which provides insight into several different questions regarding variation in environmental exposures and the association between environmental exposures and chronic diseases. The authors have done an incredibly thorough investigation into not only what factors are associated with differential environmental exposures in a large-scale cohort but also investigated both capturing single chemical exposures and mixtures of chemical exposures. This has resulted in a huge amount of information to summarize, and the authors do a wonderful job of summarizing many analyses to reach their conclusions. However, with this large amount of analysis done more detail needs to be provided to clarify what exactly was done and how it was done in order to provide additional context both to the reader and for the authors to aid in describing the strengths and limitations of this work.

Major Comments:

- The authors do provide detail in terms of the sampling design and do ensure they are capturing variability across epidemiological factors and disease status. However, no mention is made of how these individuals were recruited if the normal patients and chronic disease patients were recruited using the same process. For example, individuals recruited from a hospital, or a medical clinic – which seems to perhaps be the case given these cases are referred to as patients - might be a different group of people than those recruited from community-based sampling. Additionally, ranges of 243 to 1,813 for each disease state seems pretty notable, was that based on availability of cases, or was this intentionally done to capture more variability in different diseases? Alternatively, in the methods, the authors state they used an R package to generate the propensity score matched sample for each disease state, so perhaps this wasn't done in terms of selection of the sample but based only on the analysis regarding the disease states? More clarification is needed.

- In addition, given the sample size differences, one can easily see from Supplementary Table 6 and 7 that there is a higher number of those with hyperlipidemia and hyperLDL-C for example - and similarly a higher number of controls for these to ensure 1 to 1 propensity score matching than several other outcomes. As some disease states only have 243 cases the the authors have different degrees of statistical power which correspond to these different sample sizes. Therefore, the authors emphasis on in their abstract that “hyperlipidemia was associated with the maximum number of exposures” and in the results that this proves “that they are most affected by environmental chemicals” is not true but could be an artifact due to statistical power. Softening of this language and clarifying the sample size differences – which to some degree seem to be highlighted in Supplementary Figure 8 is imperative for interpretation of these results.

- Given the range in environmental exposure especially by region and age within this population in general, I'm a bit confused as to what value it adds to compare geometric means in concentration levels in human blood from 8 countries of various age ranges. I'm not as familiar with this literature but from googling several of these, studies like CHMS and KoNEHS are aiming at being nationally representative. In that case, are the authors hypothesizing that these are all nationally representative? And in the case of their study, depending on how the sampling was done, given in Table 1, does the geometric mean capture a nationally representative estimate given the heterogeneity by region? Additionally, are these other studies using platforms that are directly comparable to the platforms used in these other studies?

Or for example, while the authors state the variation in measures due to sampling month is explained by region, they do suggest that a broader sampling time frame might result in more variation, would this impact direct comparability of measures across this study?

Minor comments:

- The authors state that “74 exposures were found to have 50% higher detection frequency in the serum samples” but then the Supplementary Figure 1 mentions these were based on “highest detection frequency is more than 30% in 15 provinces.” I’m not as familiar with this language, but it appears they only moved forward 74 exposures which were detected with some degree of frequency within each province-specific sample? Is this the case? If they had to be detected among 30% of the sample within each province? Or 50% of the sample? Again, I am not as familiar with this type of data, but was any exploration done among those not high frequency exposures? For example, to see whether these were identified or not, more of detected yes or no in these areas?

- The authors in Figure 2B, have white to denote undetected, but in the sample, they state that they only sampled from 15 provinces. It would appear in these provinces, it was not measured. Undetected is different than unmeasured? Could more clarification be added to this figure?

- It appears when the authors reference “merge another one disease” this means, participants who have exactly one other disease, among the 12 listed. This I think is not clear from the language “merge another one disease” could they clarify that in language used?

- The authors do not classify in Figure 4, what measure of exposure these odds ratios are for? In the methods log₁₀ transformation is discussed but no mention of the z-score scaling method that were mentioned in the legend for Figure 3? If so, this needs to be mentioned ideally in both the Methods and the legend.

- Additionally, for all figures – it could be helpful to have the legends on the same scale given the focus on comparison across models. For example, in Figure 3 and 4, having the y axis of z-score concentration and the x axis of log₂ fold change be similar across all figures would allow for a better understanding of the scale of differences across age and education and similar across gender and alcohol use.

- On Figure 5C, I’m struggling with the authors statement “no significant gender differences were found in exposure risk to hyperlipidemia, but almost all exposures presented higher risk for metabolic syndrome in men.” How were gender differences tested? I could have missed this but did not see this specified in the Methods Section. Were only those which were significant in men or women added in Figure 5C? Also were none similar for men and women, or were only effects where there were differences added to Figure 5C? Also, it seems some were higher in men, and some were higher in women. And it seems there is no variation in terms of the effect sizes estimated in Figure 5C? Were all effect sizes > 0.6, or < -0.6 given the bold red and bold blue colors?

- It is a bit unclear what the authors mean by “Since hyperlipidemia, hypertension, and metabolic syndrome were determined by multiple clinical indicators, related diseases were analyzed together.” From Supplementary Figure 9, it appears 9 continuous disease parameters and 12 classified outcomes were analyzed along with hyperlipidemia and metabolic syndrome. From this Figure, it looks like this

seems to be a binary logistic regression so did they categorize this as 1 if the participant had either hyperlipidemia and metabolomic syndrome? I'm just unclear what this separate analysis truly was from this figure other than stratification by age and gender. Additionally, it appears hypertension was not grouped in this figure with hyperlipidemia and metabolomic syndrome so what was this in reference to?

- The authors focus on the overlapping chemicals in WQS, q g-comp and BKMR. It appears from their notes for example for hyperuricemia, 12, 10 and 13 preferred chemicals were obtained from the 3 models, but they report an overlap of 11. Is overlap defined as in at least two models? Could they clarify.

- While I do agree with the authors in general that their results suggest the mixture effect model is better at identifying higher risk individuals than individual compounds alone, I am a bit unclear with some wording and rationale for this statement. Specifically, the authors state the odds ratios from the reincorporated WQS and q g-comp models were "found to be significantly higher for mixture exposure than for single chemicals" but the estimates for example for the WQS model seem to be similar in magnitude to the MCHP and the PFHpS? Was there another type of testing that was done to establish statistically significantly higher? Or was it for some but not all? Additionally, the authors make the statement that "risk relationships of chemicals identified only by mixture effect models were almost non-monotonic which may be the reason why they cannot be identified by traditional single exposure models." However, there are single exposure models which can account for non-monotonic models – although this could as the authors suggest be the reason for why the single exposure models they employed did not detect them.

Reviewer #3 (Remarks to the Author):

The research provides comprehensive insights into the residue levels and exposure characteristics of environmental chemicals in human serum. They identify human serum exposures and their specific combinations that are associated with major chronic disease outcomes. Although it is innovative to a certain extent, but the analysis is simple and descriptive, not enough to support the author's research purpose and thus can't be published in such a renowned journal. There are mainly the following problems:

1. The concentration of each environmental chemicals should be provided in the main results section. What are the correlations between environmental chemicals ?
2. Did the authors adjust the medical history of each participants in the statistical analysis?
3. Figure 6l, why chooses the lag10 concentration?
4. Did the authors adjust the air pollutants (e.g. PM2.5?) and meteorological conditions such as temperature in the statistical analysis.
5. The analysis and related results of the research are relatively weak. I doubt whether it's possible to quantify the health effects of environmental chemicals, or to explore the degree of difference in the effects of environmental chemicals on various physiological processes.

Response to review results

REVIEWER COMMENTS

Reviewer #1 (Remarks to the Author):

- What are the noteworthy results?
 - o Within this study, the authors characterized a subset of the serum exposome (267 chemicals) for a study cohort spanning across China (n = 5696) via targeted GC-MS/MS and LC-MS/MS analyses. This include quantitation results with concentration levels for chemicals observed. The study cohort also included demographic data and health data, which allowed authors to perform statistical analyses in order to identify potential indicators (i.e. sources) of chemical exposure, as well as identify specific linkages between the presence of chemical and adverse health indicators as well as disease outcomes. This represents a compelling data set with enough size and diversity to identify compelling associations that may ultimately inform where chemical exposures are coming from, and which ones are concerning.

Response: Thank you for your good words for our work. Based on your suggestion we have refined the most significant findings in our research. To our knowledge, this study represents the largest research to date in terms of chemical coverage and population size. Based on this dataset, a greater number of compelling associations can be found between exposure and disease.

As you mentioned, the notable results of this study include three aspects:

1. Quantification of multiple categories of chemicals in human blood: Two methods were used to determine the detection frequency and concentration levels of 267 chemicals in a large-scale population. These results can reveal the exposure levels of different chemicals in the Chinese population and provide fundamental data for subsequent health effect studies. Specifically, we identified 74 chemicals with a high frequency detected in the human serum, including organochlorine pesticides (OCPs), organophosphorus pesticides (OPPs), polycyclic aromatic hydrocarbons (PAHs), polychlorinated biphenyls (PCBs), per- and polyfluoroalkyl substances (PFASs), as well as other drugs and pollutants.
2. Potential indicators of chemical exposure: Combined with comprehensive statistical analysis of basic epidemiological factors, it is able to understand the

potential source of chemical substances that the individuals have been/are being exposed to or accumulated. Specifically, we found that regions had the greatest influence on human exposure, followed by age. Additionally, we conducted a detailed analysis of the relationship between exposure levels and each epidemiological factor to discover patterns of variation.

3. Compelling association between exposures and chronic diseases: Through statistical analysis of various chronic disease indicators and outcomes, the associations between the presence of chemical substances and adverse health indicators were explored at multilevel. These results can provide clues regarding the potential impacts of specific chemicals on human health. Specifically, we found that hyperlipidemia was associated with the largest number of chemicals. OCPs and PFASs were associated with multiple chronic diseases, and residue levels of hexachlorobenzene, perfluorooctanoic acid (PFOA) and perfluorooctanesulfonate (PFOS) exceeded the safe ranges in some populations.

In summary, this study provides a large-scale, diverse, and rich dataset that offers powerful insights into understanding the sources of chemical exposure and their potential impacts on human health. By analyzing chemical concentration levels, potential determinants of exposure, and associations with chronic diseases, we contribute valuable information to the field.

- Will the work be of significance to the field and related fields? How does it compare to the established literature? If the work is not original, please provide relevant references.

Response: This study holds significant importance for both exposome field and environmental health research field. In existing exposome research, it is lacking of large-scale population-based cohort studies based on internal exposure monitoring, and there were scarce data regarding the health effects of internal exposure to chemicals. In terms of both the coverage of monitored chemicals and the scale of studied sample size, this study addressed the mentioned gap by constructing a high coverage quantitative method and conducting an analysis in a large-scale chronic disease cohort. It provides data of various chemical levels in human serum, exposure characteristics, and associations with chronic diseases, providing valuable resources for future research. Moreover, similarly to genomics, proteomics, and metabolomics, exposomics is an essential component for multi-level and comprehensive understanding of human health

and disease. This study develops a paradigm which contributes to an upgraded characterization of human exposome study at internal exposure level. Thus, it lays the foundation for multi-omics to jointly elucidate disease mechanisms, which is the future research trend and hotspot.

Regarding the question that “How does it compare to the established literature?”, to further illustrate the strengths and differences of this study compared to others, we have conducted a review of relevant articles on the association between environmental chemicals in human blood and chronic diseases. We have added the relevant results into the revised manuscript, and the specific method is as follows: the key words “(((human serum chemical contaminations OR serum exposures associated with (hypertension OR hyperlipidemia hyperuricemia OR diabetes OR obesity)) NOT (Acute poisoning incidents)) NOT (occupational exposure)) NOT (animal study)” were searched in PubChem yielding 2,077 articles. After manual check to find the direct correlation between chemical monitoring and chronic diseases, 275 articles were defined. The information of each article including method coverage, sample cohorts/sample size and association results has been summarized in **Supplementary Table 12**.

267 chemicals studied in this work have the highest coverage among all relevant studies. Regarding sample size, only 20 out of the 275 studies had sample sizes greater than or comparable to our study ($n > 5000$). These cohorts were from National Health and Nutrition Examination Survey (NHANES) (6 articles), the C8 Health Project (8 articles), the Canadian Health Measures Survey (CHMS) (1 article), the Italy health surveillance program (2 articles), and Korean National Environmental Health Survey (KoNEHS) (3 articles). However, the number of chemicals monitored in above studies with large-scale cohort ranged from 1 to 12, which is significantly lower than those in our study. Furthermore, there has been currently no large-scale exposome study targeting Chinese populations, hence, this study effectively fills this gap.

Regarding the originality of this study, overall, this study is original in terms of detection methods, cohort, and statistical analysis. 1) For the analytical methods, we recently established GC-MS/MS and LC-MS/MS quantitative methods, but, here we have expanded the coverage of monitored chemicals and applied them for the first time to a large-scale population cohort; 2) This study firstly provides a comprehensive description of the exposure characteristics of large samples in this Chinese cohort (Chinese Nutrition and Health Survey 2015); 3) The results of statistical analysis, including single exposure and mixed exposure effects, based on high coverage

quantitative exposures were reported for the first time. It is worth noting that our aim is to provide a comprehensive serum exposome atlas as a resource for future researches. This study is also, to our knowledge, the first exposome atlas analysis.

o Exposomics is increasingly of importance as it has been found that many of these chemical exposures encountered are related with the onset of disease. With that said, it is still being established how best to utilize/integrate chemical results with supporting metadata in order to identify when an observed chemical/chemical level is significant. The authors in this study use a nation-wide cohort to characterize part of the exposome in order to have a diversity in study population that can be utilized when perform statistical analyses on chemical results.

Response: Thank you for your positive evaluation of the significance of our research. We agree with that even though the general hazard of environmental chemicals is known, the specific health effects of exposures remain unclear, particularly in cases of low-dose long-term exposure. Therefore, association studies based on real world population-level observational data are very necessary. To better achieve this goal, we have made efforts in terms of chemical coverage, sample size, and statistical analysis methods.

For chemical coverage, we employed two high-coverage methods to detect 267 chemicals in human serum, which represents the highest coverage among relevant studies. For sample size, we included a nationwide cohort in 15 provinces in China to ensure population diversity and sample size for bridging the gap of related studies in China. For statistical analyses, due to the broad coverage and large sample size of chemicals, we discovered numerous robust associations. We extensively analyzed data related to both the determinants of chemical exposures and their relationships with chronic disease risks, providing comprehensive resources for future studies.

o A limitation of this study is that the "exposome", or the totality of chemical exposures an individual encounters, is broad to the point of being currently primarily undefined, and in efforts of characterizing it most exposomics studies utilize non-targeted analysis/suspect screening with large libraries of chemicals. This study performed targeted analyses with 267 chemicals, which allowed for defined chemical results with concentration levels but also is very limited in scope when determining the "exposome" of a population.

Response: We agree with that the chemicals studied in our research represent only a portion of the exposome, since comprehensive quantification of the entire exposome is challenging. Generally high-resolution MS is used to screen the potential exposome with a high coverage, but the sensitivity and quantitation accuracy are not so good as QQQ MS. Based on our previous work and literatures we defined 267 important serum chemicals, and develop accuracy quantitation methods to measure them by using QQQ MS. To the best of our knowledge this is the largest number of chemicals quantified in blood. We have made the utmost effort to characterize the "blood exposome" to reveal the human actual exposure scenario, although there is still a gap between the current coverage and full exposome. Our study has significantly improved the analysis coverage of chemicals compared to literatures (where most studies only focused on a single class of chemicals). Please refer to **Supplementary Table 12** for detailed information.

o Also, the criteria for the selection of the 267 chemicals included previous associations with health risks according to IRIS/IRAC, as such, the scope of this study was not about identifying new chemicals of concern from a public health standpoint, but further characterizing potential associations of chemicals with already known potential hazard. This work still however represents a large pool of data from which the authors have mined some compelling associations, and from which potentially others could benefit from analyzing statistically.

Response: We completely agree that our results are helpful for other persons.

Although we know general health risks of some chemicals, the potential associations between risk substances and chronic diseases are not clear. Their concentrations in body are also not clear. The knowledge of the health effects of long-term, low-dose exposures in large-scale populations remains limited. Our study explored the health effects of these chemicals in real-world exposure scenarios, some chemicals, such as fipronil sulfone, acesulfame and cyclamic acid were rarely reported in previous human biomonitoring study.

o This work represents quantitative data for 267 chemicals. Have these concentration levels been compared with tox data available for these chemicals in order to determine if there was an overlap between the levels observed, and the levels that would be significant from a health effect?

Response: In the original manuscript, we did not compare the monitored concentrations with toxicity data due to the facts that toxicity data were derived from animal/cellular experiments and cannot be directly compared to human exposure levels. However, inspired by your suggestion, we utilize reported human exposure guidance values from the literatures to link the observed doses with tox data available and determine the impact of exposure doses on health.

We have collected available exposure guidance values¹ including 1) biomonitoring equivalents (BE) values, which are population reference values proposed by the US Environmental Protection Agency (EPA) for rapidly characterizing biomonitoring levels exceeding values considered acceptable for exposure; 2) Human Biomonitoring II (HBM II), which establishes action levels for potentially associated adverse health effects established by the German HBM Commission; 3) Biomonitoring Guidance Values (BGV), which is the reference dose of human serum for causing peripheral and central nervous system toxicity; and 4) Reference Dose (RfD), which represents reported human serum reference doses that could potentially cause hepatotoxicity. These values were derived through toxicological data, pharmacokinetic model, and epidemiological risk determination, and exceeding the reference values indicates potential health risks^{2,3}.

Only 8 chemicals had blood reference values (BE, HBM II or BGV), and 11 PFASs had RfD values (see **Supplementary Table 10**). Based on the hazard quotients (HQs), we have evaluated whether the population exposure levels in our study posed health risks (**Supplementary Figure 19**). We found that most exposure levels of PFASs were within the safe range compared to the RfD, except PFOA, PFOS, PFNA, and PFUnDA with approaching or exceeding risk concentrations in a few individuals (**Supplementary Figure 19 A**). In comparison to the BE, HBM II and BGV, we discovered that many individuals had the exceeded exposure guidance values for HCB, PFOA, and PFOS, suggesting potential health risks (**Supplementary Figure 19 B**). Relevant descriptions have been given in the Results section “Risk analysis of single exposure to chronic diseases” of the main text, in the third paragraph.

- Does the work support the conclusions and claims, or is additional evidence needed?
- o The authors have described work that communicates the observation of 267 chemicals via targeted methods along with statistical associations with potential chemical source paths as well as potential health risks/disease outcomes. There are some questions

regarding the acquisition of the targeted data (accuracy, batch effects, internal standards- see below) that must be addressed in order to determine if the results are appropriate for the conclusions drawn.

Response: To ensure data quality to support our conclusions, we first evaluated the reliability of the quantitative methods (GC-MS/MS, LC-MS/MS) in terms of linearity, accuracy, and precision (see **Supplementary Tables 5 and 6**). Then quantitative curves based on internal standard correction and quality control samples were run in each batch, the process of internal standard correction was described in the "Quality Control and Assurance" section of the Methods. Finally, the stability was assessed by QC samples running through the large-scale samples (see **Supplementary Figure 4A, B**).

Due to your suggestion, in the revised manuscript, we have supplemented additional contents to further provide evidence of the reliability for our methods and results. Specifically, this includes the description of calibration curves for each batch (see **Supplementary Figure 2, Supplementary Tables 3, 4**), assessment of batch effects (see **Supplementary Figure 3**), description of quantitative accuracy (see **Supplementary Figure 4C, D**), and sample randomization (see **Supplementary Figure 5**).

Regarding your questions about accuracy, batch effects, and internal standards involved in the targeted data acquisition process, we provide a brief explanation below:

- **Accuracy:**

We ensured the accuracy of our quantification through measurements at low, medium, and high concentrations for each calibration curve (see **Supplementary Figure 2, Supplementary Tables 3 and 4**). Additionally, we assessed the absolute concentrations of quality control samples to demonstrate the accuracy of the quantification method (see **Supplementary Figure 4C, D**).

- **Batch effects:**

To address batch effects, we performing batch-specific calibration curve quantification combined with internal standard correction. These steps helped minimize batch effects during the data acquisition process (see **Supplementary Figure 3**).

- **Internal Standards:**

We carefully selected 27 internal standards that cover all monitored chemical categories. Following the principle of minimizing sampling relative standard deviation (RSD), each chemical was assigned an appropriate internal standard with minimum RSD in QCs (**Supplementary Tables 14, 15**). By applying internal standard correction based

on the assigned internal standards, batch effects are significantly reduced during analysis (see **Supplementary Figure 3C-F**).

These strategies were employed to ensure the accuracy, minimize batch effects, and optimize the use of internal standards in our targeted data acquisition process. For more specific answers, please refer to the detailed questions below.

- Are there any flaws in the data analysis, interpretation and conclusions? Do these prohibit publication or require revision?

Response: This study encountered the common issues in exposomics research, although we try our best to solve it. Firstly, despite our considerable efforts in expanding the coverage of chemicals, we selected only 267 chemicals from the vast exposome. We have already pointed out this limitation in the manuscript. Secondly, there may be unmeasured confounding factors that could influence the results. To address this concern, we have made improvements in the revised manuscript by considering nine confounding factors and additionally adjusting confounders of disease history and environmental factors (including air pollution and meteorological conditions). The results showed negligible impact on the outcomes (**Supplementary Figure 14**). Lastly, the original manuscript focused solely on the analysis of chemicals detected at high frequencies, which limited the analysis of low-frequency exposures. While this issue has been addressed in the current review process (**Supplementary Figure 15, 16**).

o The authors selected 74 exposure chemicals from the 267 initially analyzed because these were more prevalent across the study population (detected in >50% of samples). While this may down-select for chemicals with greater occurrences and potentially greater statistical significance, this seems unnecessarily conservative and restrictive for determining links between chemicals, exposure sources and health effects. While it is true that a greater presence of exposure across a population could be more significant if there are health risks associated with that exposure, it does not necessarily mean that exposure is of greater individual health risk. There is potential that the 197 chemicals not included with analyses with demographic data and health effects may actually identify some very meaningful exposure sources and/or disease linkages of even greater importance than the more prevalent 74 chemicals. If possible, it would be recommended that the authors include the 197 chemicals in their statistical analyses performed.

Response: Thank you for your suggestion. Considering the issue of low statistical power due to the limited number of effective cases for chemicals detected at low frequencies, we did not include the statistical analyses of low-frequency chemicals in the original manuscript. Based on your suggestion, we have now included the analysis of 193 low-frequency chemicals using binary logistic regression to explore their associations with each disease (see **Supplementary Figures 15, 16**).

We indeed found some new associations between low-frequency chemicals and diseases. Among these newly identified significant associations, most chemicals were positively associated with increased risk of chronic diseases, particularly in metabolic syndrome, obesity, diabetes, and hyperuricemia. Compared with the previous results, the significance of the associations between low-frequency chemicals and chronic diseases is weaker, and the confidence intervals are wider. This highlights the importance of cautious interpretation of these associations. Therefore, in the main text and key conclusions, we have only presented the association results between high frequently detected chemicals and chronic diseases.

• Is the methodology sound? Does the work meet the expected standards in your field?

Response: Analytical characterization data proves that our method is robust. To further demonstrate the reliability of our methods, we have supplemented results in terms of calibration curves, internal standard selection, batch effect calibration, QC accuracy evaluation, and sample randomization. Please refer to the specific answers below for more detailed information.

o Typically for targeted analyses, calibration standards bracket the analytical run rather than solely being run at the beginning as was done in this study. While QC samples were run throughout the analytical run for verification of instrument stability, this has the potential to skew results for samples acquired in the beginning vs. end of a run, which is particularly concerning if no sample randomization was performed.

Response: You have some misunderstandings to our calibration standards. In our study, calibration curves were inserted throughout the entire sample analysis process, rather than just run at the beginning. The ambiguity may have arisen from the statement: "A separate standard curve is run for each batch of samples before analysis." However, it is important to note that this study included a total of 49 batches (29 for GC-MS/MS and 20 for LC-MS/MS), bracketing the entire sample analysis process. To avoid any

confusion, we have added the corresponding statement in the manuscript to clarify that "Multiple calibration curves were inserted in the whole sample analysis process, and each independent calibration curve was run for each batch of samples. For GC platform each batch consisted of 158 real samples and 21 QC samples; for LC platform each batch contained 264 samples and 24 QC samples. Therefore, there were 29 and 20 independent calibration curves for the GC-MS/MS and LC-MS/MS platforms, respectively, bracketing the entire sample analysis process" (See method section "Quality control and assurance"). Additionally, we evaluated the methodological performance of each calibration curve and found that the accuracy at low, medium, and high concentrations was within the range of 80% to 120% for almost every calibration curve (see **Supplementary Figure 2, Supplementary Tables 3, 4**).

In order to minimize batch effects, we performed quantification using the calibration curve combined with internal standard correction for each batch, evaluated the effectiveness of this quantification and correction method through QC assessment. In the revised manuscript, we have taken LC-MS/MS data as example adding a comparison of the effects of raw signals, batch-specific quantification, and batch-specific quantification combined with internal standard correction (see **Supplementary Figure 3**). It can be observed that the batch-specific quantification and internal standard correction gradually reduce batch effects compared to the raw signals, and batch effects are almost eliminated in the final results (see **Supplementary Figure 3 C-F**). Furthermore, we have demonstrated the stability and accuracy of the method after batch-specific quantification and internal standard correction (see **Supplementary Figure 4**). Lastly, our samples were randomized, specifically by randomizing the order of samples according to each disease (see **Supplementary Figure 5**).

In conclusion, we used calibration curves, quality control samples, and internal standards that bracket the entire sample analysis process to minimize batch effects and ensure the stability and reliability of the method for sample analysis.

o Additionally, labelled internal standards can be used in targeted analyses to correct for matrix effects and instrument variability throughout the run. In order to most accurately correct for these effects, the labelled internal standards are usually labelled versions of the target analytes but in this study a set of labelled non-target analytes were chosen as internal standards, which the specific IS compounds inducing the lowest variability being selected for use as a given analyte's IS. This has potential to affect

results, particularly for multiple batches of analytical runs.

Response: In total 27 internal standards used are isotopes of the target analytes detected in this study. When selecting internal standards, the chemical categories were considered, and almost every category of analytes had 1-2 isotopic internal standards available (**Supplementary Table 13**). Due to the large number of substances being monitored, it was challenging to add corresponding isotopic internal standards for each analyte. Therefore, following the principle of the minimum relative standard deviation (RSD) in QCs, we assigned an internal standard to each chemical. The selected isotope internal standards for most substances correspond or were similar to the analytes. (See **Supplementary Tables 14, 15**).

By evaluating the stability of results before and after internal standard correction using quality control samples (See **Supplementary Figure 3**), it can be demonstrated that internal standard correction effectively reduces batch effects and ensures the reliability of the results.

o Analytical QC's were used to validate each analytical run. Typically these are used to assess accuracy and precision of the run. The methodology describes a criterion of <30% RSD for the QCs but makes no mention of the accuracy of the QC's. Again, this is particularly significant considering the fact that the calibration standards were run only at the beginning of the run; instrument drift throughout a run could potentially result in inaccurate (but precise) results that would pass the criteria used in this study.

Response: We sincerely apologize for leading to the misunderstanding of the statement that the calibration standards were run only at the beginning. In fact, 29 and 20 calibration curves were carried out on the GC-MS/MS and LC-MS/MS platforms, respectively, which spanned the entire analysis process to minimize the signal drift in different batches. We have also compared the original QC peak area signals with batch-specific linear quantification to assess batch effects (see **Supplementary Figure 3 A-D**), further emphasizing the importance of calibration curves that span the sample analysis.

To further demonstrate the accuracy and precision of the data we collected, except for the mentioned precision of QCs (see **Supplementary Figure 4A, B**) in the initial version. We further evaluate the accuracy, 98% and 79% of the exposures monitored by GC-MS/MS and LC-MS/MS platforms met the requirement of accuracy between 80% to 120% in QCs, indicating good data accuracy (see **Supplementary Figure 4C, D**).

Furthermore, the accuracies for each substance in calibration curves at low, medium, and high spiked concentrations have been calculated to fall within 80% ~ 120% (**Supplementary Figure 2, Supplementary Tables 3, 4**).

- Is there enough detail provided in the methods for the work to be reproduced?

Response: The sufficient details regarding sample randomization and batch effect correction were supplemented in the revised manuscript to make the work be reproduced.

- o It is unclear whether there was any sort of sample randomization performed when performing the MS analyses (see above comment on calibration standard bracketing)

Response: We confirm that all samples were randomized according to disease (see **Supplementary Figure 5**). Furthermore, the utilization of multiple calibration curves' quantification combined with internal standard correction methods greatly eliminated batch effects arising from mass spectrometry analysis. Therefore, the various association results obtained based on quantitative data of human internal exposure are reliable.

- o It is unclear if any batch effects were observed in the different analytical runs performed, and if so, if any batch correction was applied. This is of significance for such a large study run across multiple sampling times and analytical runs. Particularly considering for example, regions and sampling time were found to have the largest variations in exposures, but if they were run at different times batch effects may have contributed to these variations

Response: Batch effects during the large-scale sample runs were observed from original peak area of QC samples (see **Supplementary Figure 3A, B**). However, after performing quantification using multiple batch-specific calibration curves, the batch effects reduced (see **Supplementary Figure 3C, D**). Furthermore, after applying internal standard correction in multiple batch-specific calibration curves, the batch effects were minimized (see **Supplementary Figure 3E, F**) and could be neglected.

General suggestions:

- It would be recommended for the authors to revise some of the text as some of the language used may not accurately represent what is meant to be communicated.

Primarily, there are two areas:

- o References to the exposome: the exposome represents the totality of chemical exposures that an individual encounters, which is a very broad scope of chemicals. It has been referenced in the text as being characterized by this study, when in fact it is only characterizing a subset or portion of the exposome as it only monitors 267 chemicals. For example, see lines 125-126 in the text: “Then, the human serum exposome including 267 chemicals was comprehensively determined by GC-MS/MS and LC-MS/MS platforms”. Consider reviewing references to the exposome in the text to reflect that this study is only characterizing a portion of the exposome.

Response: Thank you for your suggestion. We have revised the sentence to "Then, a portion of the human serum exposome including 267 chemicals was comprehensively determined by GC-MS/MS and LC-MS/MS platforms." Additionally, we have added the determinative phrase "a portion of" to other mentions of the exposome in the manuscript.

- o The authors include disclaimer text at the end of the discussion indicating that this study does not provide causality of chemicals on adverse outcomes. However there are references in the text in which the language reflects causality, i.e. lines 251-253: “Specifically, OCPs and PCBs increase the risk of hypertension, diabetes, metabolic syndrome and obesity. PFASs increase the risk of hyperlipidemia, metabolic syndrome, diabetes, and hyperuricemia”. It would be suggested that the author review the text (primarily results and discussion) such that this language is updated such that it does not suggest causality.

Response: Based on your suggestion, we have revised the sentence to “Specifically, OCPs and PCBs were associated with the increase risk of hypertension, diabetes, metabolic syndrome and obesity. PFASs were associated with the increased risk of hyperlipidemia, metabolic syndrome, diabetes, and hyperuricemia”. Additionally, after careful review of the full text we also changed the similar descriptions in result section “Risk analysis of single exposure to chronic diseases”.

- This study represents a very compelling collection of data of which the authors have done statistical analyses to identify risk/health associations, but would also serve others by being available to other researchers. If the analytical data (267 chemicals across

5696 individuals) along with demographic data were provided in supplemental, while this would be a large table it would provide value to other researchers.

Response: We have provided the concentration levels of 267 chemicals in human serum in **Table 2 and Supplementary Table 2**. Due to our institutional policy regarding human biomonitoring, individual data are not permitted to be publicly shared online. However, we are able to provide the data with peer scientists who make official request for research purposes after their application is approved by both the corresponding and the sample collection institutes.

Lines 190-191: Please provide literature reference for reported values of PFAS in Yangtze.

Response: Here, we aimed to compare PFASs among the 15 provinces population based on concentrations observed in this study, we found that “People in Yangtze River Delta have the higher serum PFASs especially in Shanghai and Jiangsu.” This result does not relate to other literature reference for reported values of PFAS in the Yangtze region, so we do not include literature-reported data in this section.

Lines 200-202: Please provide references/sources of these literature values.

Response: We have already added the references as **Supplementary Table 7**, which presented specific data and relevant reference information for these publications.

Reviewer #2 (Remarks to the Author):

This is a really interesting paper which provides insight into several different questions regarding variation in environmental exposures and the association between environmental exposures and chronic diseases. The authors have done an incredibly thorough investigation into not only what factors are associated with differential environmental exposures in a large-scale cohort but also investigated both capturing single chemical exposures and mixtures of chemical exposures. This has resulted in a huge amount of information to summarize, and the authors do a wonderful job of summarizing many analyses to reach their conclusions. However, with this large amount of analysis done more detail needs to be provided to clarify what exactly was done and how it was done in order to provide additional context both to the reader and

for the authors to aid in describing the strengths and limitations of this work.

Major Comments:

- The authors do provide detail in terms of the sampling design and do ensure they are capturing variability across epidemiological factors and disease status. However, no mention is made of how these individuals were recruited if the normal patients and chronic disease patients were recruited using the same process. For example, individuals recruited from a hospital, or a medical clinic – which seems to perhaps be the case given these cases are referred to as patients - might be a different group of people than those recruited from community-based sampling. Additionally, ranges of 243 to 1,813 for each disease state seems pretty notable, was that based on availability of cases, or was this intentionally done to capture more variability in different diseases? Alternatively, in the methods, the authors state they used an R package to generate the propensity score matched sample for each disease state, so perhaps this wasn't done in terms of selection of the sample but based only on the analysis regarding the disease states? More clarification is needed.

Response: Thanks for your generally positive feedback and the important comments. We have also tried our best to enhance this manuscript based on your suggestion by supplementing relevant contents.

As for the detail of the sampling design, we have added information about the study population in the methods section as follow:

Study population and epidemiological information

A cross-sectional study containing 5696 subjects including 2141 healthy persons and 3555 patients was conducted in 15 provinces of China. 12 chronic disease outcomes had sample size ranging from 243 to 1813, they are diabetes, hyperuricemia, obesity, hypercholesterolemia, hypertriglyceridemia, metabolic syndrome, hyper diastolic blood pressure, hyper systolic blood pressure, abdominal obesity, hypertension, hyper low density lipoprotein cholesterol and hyperlipidemia (Supplementary Fig. 12B). The study population for this research is a subset of the cohort recruited from the China Nutrition and Health Survey in 2015. The samples covered different regions, genders, age groups, and various chronic diseases. The details of the China Nutrition and Health Survey have been described in previous studies^{4,5}. In brief, the cohort was selected using a multi-stage random cluster design, including 360 communities from 15 provinces with varying income levels. Then, 20 families were randomly selected

from each community to participate in the study. Well-trained researchers collected epidemiological data and blood samples, and clinical parameters related to chronic diseases were determined through biochemical assays of the blood samples.

The epidemiological factors were gathered by questionnaire, including gender, age, sampling location, sampling time, education level, income level, marital status, cigarette smoking, alcohol drinking and medical histories. Moreover, 9 chronic disease-related clinical parameters were determined including uric acid, glycosylated hemoglobin, low density lipoprotein cholesterol, triglyceride, total cholesterol, systolic blood pressure, diastolic blood pressure, waistline and BMI. The detailed information is provided in Table 1. 12 chronic disease outcomes were subdivided using these 9 related clinical parameters (**Supplementary Note 1**). The cohort has conducted 10 rounds of surveys from 1989 to 2015. In the 2015 survey, 14,000 samples were collected, we randomly selected half of these samples for exposome research. After excluding samples with insufficient volume or incomplete information, a total of 5,696 samples were finally analyzed in this study. The study has been approved by the National Institute for Nutrition and Health, Chinese Center for Disease Control and Prevention (reference No. 201524), and a written informed consent was obtained from each participant before the study began.

In summary, the samples in this study were obtained through community sampling, and the disease samples were classified based on the clinical parameters tested rather than being obtained from hospitals. The sample size ranging from 243 to 1,813 for each disease is real reflection of various chronic diseases prevalence in the general population.

- In addition, given the sample sizes differences, one can easily see from Supplementary Table 6 and 7 that there is a higher number of those with hyperlipidemia and hyperLDL-C for example - and similarly a higher number of controls for these to ensure 1 to 1 propensity score matching than several other outcomes. As some disease states only have 243 cases the the authors have different degrees of statistical power which correspond to these different sample sizes. Therefore, the authors emphasis on in their abstract that “hyperlipidemia was associated with the maximum number of exposures” and in the results that this proves “that they are most affected by environmental chemicals” is not true but could be an artifact due to statistical power. Softening of this language and clarifying the sample size differences – which to some degree seem to be

highlighted in Supplementary Figure 8 is imperative for interpretation of these results. Response: We agree that sample size affects statistical power. In theory, a larger sample size could increase statistical power. It means more accurate associations can be identified. In our study, for example, the sample size for hypertension (n=1323) is only slightly less than that of hyperlipidemia (n=1813), but there are very few chemicals found to be associated with hypertension. This indicates that a larger sample size does not necessarily lead to the discovery of more associations. Nevertheless, it is undeniable that smaller sample sizes can increase the random errors.

To validate our previous conclusion that "hyperlipidemia was associated with the maximum number of exposures" we randomly selected 300 cases of hyperlipidemia for further analysis and still found the highest number of associations (see **Supporting Figure R1 for Reviewer**). Therefore, we maintain the original conclusion.

Supporting Figure R1 for Reviewer. The odds ratios of high-frequency exposure to hyperlipidemia (n = 300). OR represents odds ratio per one-unit increase in log-transformed exposure levels. Binary logistic regression model adjusted for age, gender, region, sampling time, education and income levels, marital status, smoking and drinking history and a subset of hyperlipidemia with 300 samples were used in the regression model. The position and color of diamond represent odds ratios and significant (two-sided), respectively. Grey diamonds represent no significance. Error bars represent 95% confidence interval of odds ratios.

- Given the range in environmental exposure especially by region and age within this population in general, I'm a bit confused as to what value it adds to compare geometric means in concentration levels in human blood from 8 countries of various age ranges. I'm not as familiar with this literature but from googling several of these, studies like CHMS and KoNEHS are aiming at being nationally representative. In that case, are the authors hypothesizing that these are all nationally representative? And in the case of their study, depending on how the sampling was done, given in Table 1, does the geometric mean capture a nationally representative estimate given the heterogeneity by region? Additionally, are these other studies using platforms that are directly comparable to the platforms used in these other studies? Or for example, while the authors state the variation in measures due to sampling month is explained by region, they do suggest that a broader sampling time frame might result in more variation, would this impact direct comparability of measures across this study?

Response: Our studied were collected from 15 provinces, including the southern and northern regions as well as coastal and inland areas across the country, representing 56% of the Chinese population, it provides the most representative data for chemical biomonitoring in the Chinese population to date.

To gain a global understanding of serum exposure levels, we compared the geometric mean concentrations of serum chemicals from 8 countries worldwide. We attempted to select national-level cohorts representing the average serum exposure level of the entire population from the available data. Next, we used the geometric mean of our study's data as the average serum exposure level for the Chinese population. And compared our data with those from other countries to provide new insights into global human exposure levels.

The detection data from our study and other studies were all quantified using GC-MS/MS and LC-MS/MS, although raw MS signal fluctuated during the analysis process, the absolute concentrations theoretically should be consistent across different studies, especially when the same kind of platform was selected, making the final concentration data comparable. It should be noted that a broader sampling time frame might result in more variation and reduce the comparability of results. To address this issue, we made efforts to select studies conducted around the same time as our study (2015). Additionally, we halabeled the sampling time information in the figures, and suggest readers to consider this when interpreting the results cautiously.

Minor comments:

- The authors state that “74 exposures were found to have 50% higher detection frequency in the serum samples” but then the Supplementary Figure 1 mentions these were based on “highest detection frequency is more than 30% in 15 provinces.” I’m not as familiar with this language, but it appears they only moved forward 74 exposures which were detected with some degree of frequency within each province-specific sample? Is this the case? If they had to be detected among 30% of the sample within each province? Or 50% of the sample? Again, I am not as familiar with this type of data, but was any exploration done among those not high frequency exposures? For example, to see whether these were identified or not, more of detected yes or no in these areas?

Response: We sincerely apologize for our oversight in the legend of Supplementary Figure 1. It should indeed be 50% instead of 30%, and we have corrected this mistake and added an explanation in the legend. Additionally, in the Methods section, we have indicated the criteria for the high-frequency chemicals, where 74 chemicals were identified with a highest detection frequency of more than 50% in 15 provinces. This means that these chemicals were considered high frequency if they had a detection frequency exceeding 50% in any one of provinces. The reason for such a selection was to include more chemicals and ensure statistical analysis power.

We acknowledge that there may be meaningful associations among the 193 low-frequency exposures that could have been overlooked, as reviewer #1 suggested. We conducted additional analyses (see **Supplementary Figure 15, 16**) and found that the newly included low-frequency exposures exhibited weaker significance and wider confidence intervals in their associations with chronic diseases. This suggests that we should refer to these associations with caution. As a result, we only present the association results between high frequently detected chemicals and chronic diseases in the main text and conclusions.

- The authors in Figure 2B, have white to denote undetected, but in the sample, they state that they only sampled from 15 provinces. It would appear in these provinces, it was not measured. Undetected is different than unmeasured? Could more clarification be added to this figure?

Response: Thank you for the correction. But there is no “undetected” label in Figure 2B, so I guess you are talking about Figure 3A. The white color in Figure 3A represents

“unmeasured” rather than “undetected”, and this has been corrected accordingly in the figure.

- It appears when the authors reference “merge another one disease” this means, participants who have exactly one other disease, among the 12 listed. This I think is not clear from the language “merge another one disease” could they clarify that in language used?

Response: We apologize for any confusion caused by the phrase "merge another one disease." Here, “merge another one disease” represents an individual who suffers from only two chronic diseases in total at the same time. To make it clearer we have changed it into “two diseases”.

- The authors do not classify in Figure 4, what measure of exposure these odds ratios are for? In the methods log₁₀ transformation is discussed but no mention of the z-score scaling method that were mentioned in the legend for Figure 3? If so, this needs to be mentioned ideally in both the Methods and the legend.

Response: It is our negligence not to clarify the measure of exposure these odds ratios. We have added instructions in the legend: “OR represent odds ratio per one-unit increase in log-transformed exposure levels”. For Figure 3, the data are transformed using z-scores to eliminate the influence of different units or scales and allow for comparisons among all chemicals. Figure 4, on the other hand, involves a log₁₀ transformation of the data to convert skewed data into a normal distribution, thereby reducing heteroscedasticity and improving the fitting of regression models as well as the reliability of statistical inferences. To make it clear, we have explicitly mentioned this data processing approach not only in the Methods section but also in the figure legends.

- Additionally, for all figures – it could be helpful to have the legends on the same scale given the focus on comparison across models. For example, in Figure 3 and 4, having the y axis of z-score concentration and the x axis of log₂ fold change be similar across all figures would allow for a better understanding of the scale of differences across age and education and similar across gender and alcohol use.

Response: This is a good suggestion. To facilitate comparison, we have adjusted the scales of **Figure 3C-H** and **Figure 4A-L** to be the same. Additionally, we have also

adjusted the scale of the legend of **Figure 6 A-F** and **Supplementary Figure 3, 11, 14, 15 and 19**.

- On Figure 5C, I'm struggling with the authors statement "no significant gender differences were found in exposure risk to hyperlipidemia, but almost all exposures presented higher risk for metabolic syndrome in men." How were gender differences tested? I could have missed this but did not see this specified in the Methods Section. Were only those which were significant in men or women added in Figure 5C? Also were none similar for men and women, or were only effects where there were differences added to Figure 5C? Also, it seems some were higher in men, and some were higher in women. And it seems there is no variation in terms of the effect sizes estimated in Figure 5C? Were all effect sizes > 0.6 , or < -0.6 given the bold red and bold blue colors?

Response: In the statistical analysis section (the third paragraph "Risk of disease for each individual chemical"), we have described how gender differences were addressed. However, we acknowledge that our description may have been oversimplistic and led to ambiguity. To be clear, we have added relevant details as follows: "To understand susceptibility to disease risk from exposure in different age and gender populations, samples were grouped by age and gender. Specifically, the population was divided into three age groups: 30-50 years old, 50-60 years old, and 60-80 years old. The population was also divided into two gender groups: males and females. Subsequently, propensity score matching was used to classify the data sets into disease and normal groups for each subgroup. Finally, binary logistic regression was performed to determine odds ratios (ORs) for the exposures in each group."

Significant chemicals identified in binary logistic regression models for the overall population and at least one gender group were included in Figure 5C, and corresponding descriptions have been added to the legend. You are right that Figure 5C does not reflect the variation of the effect size. This is because Figure 5C presents the results of scaled ORs, where the smaller odds ratios between males and females is scaled to -0.6 and the larger value is scaled to 0.6 . Therefore, this figure can solely depict the relative sizes of OR values between males and females but does not represent the absolute values. We have supplemented the results with a figure displaying the absolute values of ORs, further analyzing the gender differences in the correlation between exposures and disease outcomes (see **Supplementary Figure 18**) which was still concordant with our

conclusions. “no significant gender differences were found in exposure risk to hyperlipidemia, but almost all exposures presented higher risk for metabolic syndrome in men.”

- It is a bit unclear what the authors mean by “Since hyperlipidemia, hypertension, and metabolic syndrome were determined by multiple clinical indicators, related diseases were analyzed together.” From Supplementary Figure 9, it appears 9 continuous disease parameters and 12 classified outcomes were analyzed along with hyperlipidemia and metabolic syndrome. From this Figure, it looks like this seems to be a binary logistic regression so did they categorize this as 1 if the participant had either hyperlipidemia and metabolic syndrome? I’m just unclear what this separate analysis truly was from this figure other than stratification by age and gender. Additionally, it appears hypertension was not grouped in this figure with hyperlipidemia and metabolic syndrome so what was this in reference to?

Response: Thank you for bringing this concern to our attention. Our original intention was to convey that some of these 12 diseases was correlated, for example, hyperlipidemia is determined by multiple clinical indicators including hyper LDL-C, hypercholesterolemia, and hypertriglyceridemia, so we called them as related diseases of hyperlipidemia, so did hypertension.

To make it clearer, we have changed it to “Among the 12 chronic diseases, three of them, namely hyperlipidemia, hypertension, and metabolic syndrome were determined by multiple clinical parameters (**Supplementary Fig. 12C-E**). The remaining 9 chronic diseases were determined by single clinical parameters (**Supplementary Note 1**).”

The specific criteria for classification of 12 chronic disease outcomes by 9 clinical parameters.

1. Individuals with uric acid levels equal to or greater than 420 μ mol/L for males, and equal to or greater than 360 μ mol/L for females are determined to have hyperuricemia.
2. Individuals with glycated hemoglobin levels equal to or greater than 6.5% are determined to have diabetes.
3. Individuals with low-density lipoprotein cholesterol levels equal to or greater than 3.4mmol/L are determined to have hyperLDL-C.
4. Individuals with triglyceride concentrations equal to or greater than 2.3mmol/L are determined to have hypertriglyceridemia.

5. Individuals with total cholesterol levels equal to or greater than 6.2mmol/L are determined to have hypercholesterolemia.
6. Individuals with systolic blood pressure equal to or greater than 140mmHg are determined to have high systolic blood pressure.
7. Individuals with diastolic blood pressure equal to or greater than 90mmHg are determined to have high diastolic blood pressure.
8. Individuals with waist circumference equal to or greater than 90cm for males, and equal to or greater than 85cm for females are determined to have abdominal obesity.
9. Individuals with a BMI (Body Mass Index) greater than 28kg/m² are determined to be obese.
10. Hypertension is determined if either the systolic blood pressure is equal to or greater than 140mmHg, or the diastolic blood pressure is equal to or greater than 90mmHg (refer to **Supplementary Figure 12C**).
11. Hyperlipidemia is determined if any of the three clinical indicators are met: low-density lipoprotein cholesterol equal to or greater than 3.4mmol/L, triglyceride concentration equal to or greater than 2.3mmol/L, or total cholesterol equal to or greater than 6.2mmol/L (refer to **Supplementary Figure 12D**).
12. Metabolic syndrome is determined if an individual has three or more of the following four chronic diseases: abdominal obesity, hypertriglyceridemia, hypertension, and diabetes (refer to **Supplementary Figure 12E**).

To answer the question about Supplementary Figure 9 (now **Supplementary Figure 13**): we present 3 statistical analysis methods for 9 clinical indicators mentioned along with the 12 and 2 disease outcomes. They are considered as parallel and separate analysis, where multiple linear regression is used for the clinical indicators, logistic regression is used for the 12 outcomes, and stratified logistic regression is employed for the two disease outcomes (hyperlipidemia and metabolic syndrome). For better understanding, we have rearranged them to match the order they appear in the results section “Risk analysis of single exposure to chronic diseases”. Specifically, in the first paragraph, we presented in detail the results of binary logistic regression for 12 disease outcomes, followed by the validation of these results using 9 clinical parameters. In the second paragraph, we showed the results of 2 diseases stratified by gender and age.

To answer the question about why hypertension was not grouped in this figure:

only two chronic disease including hyperlipidemia and metabolic syndrome were selected in age and gender stratified analysis. The reasons for selecting these two diseases are explained in the article as follows: "Hyperlipidemia and metabolic syndrome were chosen as target diseases because of the sufficient sample size and the exposure-disease associations found above." Although there were sufficient samples of hypertension patients for age and gender stratified analysis, only a very small number of chemicals showed associations with hypertension in the exposure-disease association study. Therefore, hypertension was not included in the stratified analysis.

- The authors focus on the overlapping chemicals in WQS, q g-comp and BKMR. It appears from their notes for example for hyperuricemia, 12, 10 and 13 preferred chemicals were obtained from the 3 models, but they report an overlap of 11. Is overlap defined as in at least two models? Could they clarify.

Response: Your understanding is right. We have defined the chemicals that were selected in at least two mixed exposure models as the overlapping chemicals of key interest. To make it clear, we have highlighted them in Venn diagram in red, and also added an explanation in the text and legend of **Supplementary Figure 20E, J, O**.

- While I do agree with the authors in general that their results suggest the mixture effect model is better at identifying higher risk individuals than individual compounds alone, I am a bit unclear with some wording and rationale for this statement. Specifically, the authors state the odds ratios from the reincorporated WQS and q g-comp models were "found to be significantly higher for mixture exposure than for single chemicals" but the estimates for example for the WQS model seem to be similar in magnitude to the MCHP and the PFHpS? Was there another type of testing that was done to establish statistically significantly higher? Or was it for some but not all? Additionally, the authors make the statement that "risk relationships of chemicals identified only by mixture effect models were almost non-monotonic which may be the reason why they cannot be identified by traditional single exposure models." However, there are single exposure models which can account for non-monotonic models – although this could as the authors suggest be the reason for why the single exposure models they employed did not detect them.

Response: Thank you very much for pointing out this problem, which gave us the opportunity to investigate the reasons behind the anomalous results and make

improvements to obtain more robust findings. We apologize for the error that due to the lack of a seed and environment configuration in the original models, random results were generated. After correcting the parameters and conducting multiple tests, we obtained modified results as shown in **Figure 6, Supplementary Figures 20 and 21, Supplementary Table 11.**

In addition, we have also modified the manuscript in the results section (see the result section “Risk effects of exposure mixtures on related chronic diseases”). In summary, the adjusted models affected the selection of some overlapping key exposures, but overall, most of the results remained unchanged. The updated results better support our conclusion that " Mixtures of these priority risk chemicals were reincorporated into the WQS and q g-comp models to obtain odds ratios which were found to be significantly higher for mixture exposure than for single chemicals."

We have also rechecked the qg-comp and BKMR models and did not find unstable results as that in the WQS model. This further demonstrates the reliability of combining these three mixed exposure models to obtain results.

To answer the reviewer’s question about non-monotonic association : We agree that there are single exposure models which can account for non-monotonic models, such as generalized additive models (GAMs). Our conclusions are based on the statistical models used in this study, where the single exposure models employed, including multiple linear regression and binary logistic regression, are linear models. On the other hand, the mixed exposure models could be able to identify non-linear relationships. To avoid overstatement, we have added the qualifying statement to revise the text as follows: "The risk relationships of chemicals identified only by mixture effect models were almost non-monotonic (**Supplementary Fig. 21**), which may be the reason why they cannot be identified by traditional linear single exposure models such as linear and logistic regression model used in this study."

Reviewer #3 (Remarks to the Author):

The research provides comprehensive insights into the residue levels and exposure characteristics of environmental chemicals in human serum. They identify human serum exposures and their specific combinations that are associated with major chronic disease outcomes. Although it is innovative to a certain extent, but the analysis is simple

and descriptive, not enough to support the author's research purpose and thus can't be published in such a renowned journal. There are mainly the following problems:

1. The concentration of each environmental chemicals should be provided in the main results section. What are the correlations between environmental chemicals ?

Response: Thank you very much for your suggestion. In the original manuscript, we presented the concentrations of each environmental chemical in **Supplementary Table 2**. However, based on your recommendation, we have now included the concentration levels of high-frequency chemicals relevant to the main content of the article in **Table 2** within the main text. In addition, the correlation among the concentrations of key environmental chemicals was illustrated in **Supplementary Figure 8**. In the main text, we describe it as follows: " The strongest association was demonstrated between PFASs followed by OCPs, PCBs and PAHs (**Fig. 2E; Supplementary Fig. 8**). This suggests that chemicals in the same category are likely from similar sources of contamination. "

2. Did the authors adjust the medical history of each participants in the statistical analysis?

Response: We appreciate you for bringing this point to our attention. In the original manuscript, we did not consider adjusting for disease history as a confounding factor. Due to your suggestion, we have added the results of correcting for disease history (**Supplementary Figure 14** model 1) in the revised version. We found that after adjusting for disease history, including histories of hypertension, diabetes, myocardial infarction, stroke, and cancer, most of the associations between individual chemicals and diseases remain unchanged.

3. Figure 6I, why chooses the lag10 concentration?

Response: The concentrations of exposure variables were logarithmically transformed to convert skewed data into a normal distribution, thereby reducing heteroscedasticity and improving the reliability of statistical inferences of logistic regression models. Also, it has been reported in the literature that log 10 transformed concentrations were used in dose-risk relationship analyses⁶. Therefore, we chose the log10 concentration.

4. Did the authors adjust the air pollutants (e.g. PM2.5?) and meteorological conditions such as temperature in the statistical analysis.

Response: In our previous manuscript, we did not adequately consider the impact of confounding factors such as air pollution and meteorological conditions on the association results. Because air pollution and meteorological conditions can act as triggering factors for chronic diseases and may influence the association results, we have included additional analyses to address this issue. We obtained air pollution and meteorological data using regional and sampling time (month) information from open-source websites. The detailed method is described in the Methods section of the **Supplementary Note 2**.

“Environmental factors including air pollution and meteorological conditions were obtained using regional and sampling time (month) information from open-source websites. Specifically, for air pollution, data on three variables including Air Quality Index (AQI), PM_{2.5}, and PM₁₀ were collected from the website <https://quotsoft.net/air/>. Daily data for each region (specific to city/county) were available, and based on these data, monthly values of AQI, PM_{2.5}, and PM₁₀ were calculated for each individual in their respective region (specific to city/county) and sampling month.

For meteorological conditions, data on three temperature variables including daily maximum temperature, daily minimum temperature, and daily average temperature were collected from the website <https://www.geodoi.ac.cn/WebCn/doi.aspx?Id=3187>. Daily data for each region (specific to province) were accessible, and based on these data, monthly mean values of Daily_Maximum_Temp, Daily_Mean_Temp, and Daily_Minimum_Temp were calculated for each individual in their respective region (specific to province) and sampling month.

Finally, the obtained six environmental variables and nine epidemiological factors were treated as confounding factors and adjusted by binary logistic regression model.”

The results have been presented in **Supplementary Figure 14** (model 2) which showed that the additional adjustment for air pollution and meteorological conditions had minimal impact on the association results. This is mainly because in the case-control matching, we rigorously matched the regions and sampling time, ensuring that there were no significant differences in air pollution and meteorological conditions between the disease and control groups. This further proves the robustness of our findings.

5. The analysis and related results of the research are relatively weak. I doubt

whether it's possible to quantify the health effects of environmental chemicals, or to explore the degree of difference in the effects of environmental chemicals on various physiological processes.

Response: Thank you for your suggestion. Quantitatively describing the health effects of chemicals is indeed a meaningful research goal. We attempted to answer this question from two perspectives: what concentrations of environmental chemicals would cause health effects, and which chemicals pose a greater risk for specific diseases. Firstly, we obtained exposure guidance values from the literature to address the question of what concentrations of environmental chemicals would cause health effects. These available exposure guidance values include BE, HBM II, BGV and RfD, which are derived from toxicological data, pharmacokinetic model and epidemiological data^{2,3}. By comparing the observed chemical concentrations in this study with these exposure guidance values, we assessed the health risk of each individual in this study using a hazard quotients (HQs) approach. The results indicated that some individuals exceeded the safe range of PFOA, PFOA, and HCB (**Supplementary Figure 19 B**).

Additionally, there is currently a lack of exposure guidance values based on human biomonitoring, and different organizations have significant variations in their reference values. Using dose-response relationship curves, we established exposure guidance values for high-frequently detected chemicals related to hyperlipidemia by our data. We found that the guide levels for PFOA (3.19 ng/mL) and PFOS (5.26 ng/mL) were slightly lower compared to the literature-reported values (PFOA (10 ng/mL) and PFOS (20 ng/mL)). In addition, there are 5 chemical exposure guidance values that are not reported including beta-HCH, p,p'-DDE, PFHpS, 6:2 Cl-PFAES and MEP (**Supplementary Figure 19C**). Relevant descriptions were in the Results section “Risk analysis of single exposure to chronic diseases” of the main text, in the third paragraph.

Finally, we compared the magnitude of odds ratios among chemicals within the same disease group to determine which chemicals posed a greater risk. Taking hyperlipidemia as an example, among organochlorine pesticides, pp-DDT had the highest ratio, while among perfluorocarboxylic acids and perfluorosulfonic acids, PFNA and PFHpS had the highest ratios, respectively. This suggests that these chemicals have a greater impact on hyperlipidemia.

References

1. Nakayama, S.F. *et al.* Interpreting biomonitoring data: Introducing the international human biomonitoring (i-HBM) working group's health-based guidance value (HB2GV) dashboard. *Int. J. Hyg. Environ. Health* **247**, 114046 (2023).
2. Hays, S.M. *et al.* Guidelines for the derivation of Biomonitoring Equivalents: report from the Biomonitoring Equivalents Expert Workshop. *Regul. Toxicol. Pharmacol.* **51**, S4-15 (2008).
3. Angerer, J., *et al.* Human biomonitoring assessment values: approaches and data requirements. *Int. J. Hyg. Environ. Health* **214**, 348-360 (2011).
4. Zhang, B., Zhai, F.Y., Du, S.F. & Popkin, B.M. The China Health and Nutrition Survey, 1989-2011. *Obes Rev* **15 Suppl 1**, 2-7 (2014).
5. Su, C. *et al.* Longitudinal Association between Urbanicity and Total Dietary Fat Intake in Adults in Urbanizing China from 1991 to 2015: Findings from the CHNS. *Nutrients* **12**, 1597 (2020).
6. Wang, B. *et al.* Acrylamide exposure increases cardiovascular risk of general adult population probably by inducing oxidative stress, inflammation, and TGF-beta1: A prospective cohort study. *Environ. Int.* **164**, 107261 (2022).

REVIEWER COMMENTS

Reviewer #1 - withdrawn

Reviewer #2 (Remarks to the Author):

I want to thank the authors for their clarification and edits. There are still a few outstanding points that I'm hoping they can clarify which I think will help the paper prior to publication.

- The author's addition of the sampling design is incredibly clear and helpful. I am still however a bit perplexed then given the sampling design about what the authors mean when they say "Subsequently, propensity score matching was used to classify the dataset into disease and control groups for each subgroup." In what context were propensity scores used? Don't they just have a number of individuals given the study design with disease and a number of individuals without?

- With regards to the comment "hyperlipidemia was associated with the maximum number of exposures" I appreciate the authors taking the time to explore the robustness of their findings. I understand they feel comfortable still saying this. However, I think I'm still confused as to why this is an important finding and therefore why it becomes important to present in the abstract? It doesn't seem to be a main focus of the discussion and I think instead the authors narrow the focus on the discussion to provide important context not just about the number of significant results but rather what exposures are associated with hyperlipidemia. I know this is a minor point but if they want to keep this sentence in the abstract, can they provide more information – to me along with the sample size (even though it is robust) in combination with the idea that the number of exposures associated with any outcome will vary depending on what exposures were measured, makes it hard to interpret and understand why the authors feel this is an important finding.

- In terms of the response to comparisons between CHMS and KoNEHS and other studies, I think providing the context they describe in the paper would be crucial. Understanding the study design – which does as the authors state – include data from 15 provinces representing 56% of the Chinese population and provides the most representative data for chemical biomonitoring in the Chinese population to date – this information I think when referring to these comparisons would be helpful for the reader to understand the context of why this is a reasonable comparison.

Reviewer #3 (Remarks to the Author):

I have no further comments.

Reviewer #4 (Remarks to the Author): - replacement for Reviewer #1

Dear authors,

Please find my comments as below and my response for mass spec-related comments and questions from previous reviewers are attached. I appreciate authors' contribution to deep understanding of the environmental exposure as risk factors for chronic diseases. This study involves more than 5000 participants, investigating correlation of 267 chemical exposures and 12 chronic diseases. Overall, this study provides a rich resource for the exposome research, and the scientific content has potential impact to understand the relationship of environmental hazard and chronic diseases in east Asia.

Major points

1. From the statistics perspective, one of major concerns is that authors didn't address which value of each chemical was used for the regression models. How about the ones with low detection frequency. The chemicals with non-detectable or below the LOQ are treated as NA or zero? The statistics section doesn't contain essential and enough information to understand how the informatic analyses were performed. For example, in most of the case, only the R packages used in this study were mentioned (e.g., PCA); this is a very vague description without know which samples and/or readouts were used for PCA. Despite Suppl Note 6 described 3 models, this is just generic description of the algorithms but no related content to this study.
2. From the analytical chemistry perspective, it is not clear that authors designed one targeted method each for GCMS and LCMS to monitor 97 and 170 chemicals, respectively. Or multiple method was used. I would like to see some MS/MS spectra in supplemental materials, and the raw MS files should be accessible to public (not by request). I noticed not all chemicals have their isotopic labeled counterparts (e.g., salmeterol, fenpyroximate, etc). How can be sure analytes are real without standard and matched retention time in low-resolution mass spectrometer.
3. The description of batch correction approach is vague. First, the authors didn't define what are the batches, are those samples were multiplex bar-coded or simply their data were acquired in a short period of time? I believe the defined batch has no solid foundation. Furthermore, how exactly "batch-specific calibration curve" approach works. The median or mean values of 21 and 24 QCs in GC and LC platform, respectively, were used for standard curve construction? Often studies only use external or internal standards for normalization and batch correction. Please elaborate how the statistical works

integrated of standard curve (external standards) and a handful spike-in internal standards. I believe this is the foundation of how accurate the exposome are assessed.

Minor points

1. The title of this article is too broad not very specific addressing the exposome investigation only in east Asia (China), this could lead to race/population bias.
2. In the same regard, this study cannot be called as an exposome atlas with only 267 chemicals.
3. Figure 1 is overstuffed with many supplemental figures and doesn't contain critical information for understanding the scientific contents, for example, no X and Y axis titles. This is a more like a graph abstract not an actual figure.
4. In suppl note 4, only LC parameters but no MS ones were addressed, such as cycle time, etc.

REVIEWER COMMENTS

Reviewer #1 - withdrawn

Reviewer #2 (Remarks to the Author):

I want to thank the authors for their clarification and edits. There are a still a few outstanding points that I'm hoping they can clarify which I think will help the paper prior to publication.

- The author's addition of the sampling design is incredibly clear and helpful. I am still however a bit perplexed then given the sampling design about what the authors mean when they say "Subsequently, propensity score matching was used to classify the dataset into disease and control groups for each subgroup." In what context were propensity scores used? Don't they just have a number of individuals given the study design with disease and a number of individuals without?

Response: Thank you for the general positive comments. In this study, propensity score matching (PSM) was used in two scenarios. One was for analyzing the risk of exposures to each chronic disease, and the other was for stratified risk analysis based on gender and age (see "Risk of disease for each individual chemical" of Statistical analysis part). You are right that PSM is used to match some diseased individuals with health individuals, but these two scenarios are a little bit different. In the first scenario, taking metabolic syndrome for example, PSM, which considered all epidemiological factors and categorical variable of chronic diseases, was used to match health ones for metabolic syndrome individuals. In the second scenario which focused stratified risk analysis, all individuals were initially grouped by targeted stratification factor including age and gender, then PSM was conducted according to the other epidemiological factors and categorical variable of disease. For example, when investigating the specific risk of exposures to metabolic syndrome in males, all males are initially grouped, then PSM is performed to ensure more accurate matching of gender factors between the disease and control groups.

- With regards to the comment "hyperlipidemia was associated with the maximum number of exposures" I appreciate the authors taking the time to explore the robustness of their findings. I understand they feel comfortable still saying this. However, I think I'm still confused as to why this is an important finding and therefore why it becomes important to present in the abstract? It doesn't seem to be a main focus of the discussion and I think instead the authors narrow the focus on the discussion to provide important context not just about the number of significant results but rather what exposures are associated with hyperlipidemia. I know this is minor point but if they want to keep this sentence in the abstract, can they provide more information – to me along with the sample size (even though it robust) in combination with the idea that the number of exposures associated with any outcome will vary depending on what exposures were

measured, makes it hard to interpret and understand why the authors feel this is an important finding.

Response: We agree with your point. Our original idea was that hyperlipidemia is associated with the maximum number of exposures, which may suggest that environmental chemical exposures have the greatest impact on hyperlipidemia. Therefore, we believed this was an important finding. However, based on your suggestion, we realized that the number of associations may vary due to the different types of chemicals monitored, we should focus more on what chemicals are associated with hyperlipidemia just as what is described in the discussion section of the manuscript. As a result, we have removed this conclusion from the abstract.

- In terms of the response to comparisons between CHMS and KoNEHS and other studies, I think providing the context they describe in the paper would be crucial. Understanding the study design – which does as the authors state – include data from 15 provinces representing 56% of the Chinese population and provides the most representative data for chemical biomonitoring in the Chinese population to date – this information I think when referring these comparisons would be helpful for the reader to understand the context of why this is a reasonable comparison.

Response: Thank you for your valuable suggestion. In order to better explain the rationale behind comparing our study data with other national cohorts, we have added relevant descriptions based on your suggestion in the discussion section. The detail text is that “Additionally, the results from the cohort in this study provide the most representative data for chemical biomonitoring in the Chinese population to date. The cohort in this study covered 15 provinces, including the southern and northern regions as well as coastal and inland areas across the country, representing 56% of the Chinese population. The result showed that the level of OCPs in human blood is significantly higher in China than in other countries...”.

Reviewer #3 (Remarks to the Author):

I have no further comments.

Response: Thank you.

Reviewer #4 (Remarks to the Author): - replacement for Reviewer #1

Dear authors,

Please find my comments as below and my response for mass spec-related comments and questions from previous reviewers are attached. I appreciate authors' contribution to deep understanding of the environmental exposure as risk factors for chronic diseases. This study involves more than 5000 participants, investigating correlation of 267 chemical exposures and 12 chronic diseases. Overall, this study provides a rich resource for the exposome research, and the scientific content has potential impact to understand the relationship of environmental hazard and chronic diseases in east Asia.

Major points

1. From the statistics perspective, one of major concerns is that authors didn't address which value of each chemical was used for the regression models. How about the ones with low detection frequency. The chemicals with non-detectable or below the LOQ are treated as NA or zero? The statistics section doesn't contain essential and enough information to understand how the informatic analyses were performed. For example, in most of the case, only the R packages used in this study were mentioned (e.g., PCA); this is a very vague description without know which samples and/or readouts were used for PCA. Despite Suppl Note 6 described 3 models, this is just generic description of the algorithms but no related content to this study.

Response: Thank you for your generally positive feedback and the important comments. As for the question that "which value of each chemical was used for the regression models", we have described that "The exposure concentration levels used were log10 transformed for all of the regression analysis." in the fourth paragraph of statistical analysis section.

The concentrations of each chemical below the limit of quantification (LOQ) are set as $LOQ/\sqrt{2}$ (see the first paragraph of statistical analysis section). This method of filling in the missing concentration values is commonly used in human biomonitoring researches¹⁻³.

Due to space limitations, R package analysis process was not provided in previous version of manuscript. Following your suggestion, in order to give readers a better understanding of how data analysis is conducted, we have made it available on GitHub including the detailed code and example files for the R packages used in this study (see <https://github.com/youlei2023/ExposomeAtlas>).

2. From the analytical chemistry perspective, it is not clear that authors designed one targeted method each for GCMS and LCMS to monitor 97 and 170 chemicals, respectively. Or multiple method was used. I would like to see some MS/MS spectra in supplemental materials, and the raw MS files should be accessible to public (not by request). I noticed not all chemicals have their isotopic labeled counterparts (e.g., salmeterol, fenpyroximate, etc). How can be sure analytes are real without standard and matched retention time in low-resolution mass spectrometer.

Response: Actually, two targeted methods, which covered 97 chemicals in GC-MS/MS platform and 170 chemicals in LC-MS/MS platform, respectively, have been used in this study. To make it clearer, we have changed the description in the last paragraph of introduction part into "Then, two targeted methods covering 267 environmental chemicals usually met and reported were established to quantify a portion of the serum exposome, one is based on the gas chromatography-tandem mass spectrometry (GC-MS/MS) platform for 97 chemicals quantification, and the other is based on liquid chromatography (LC)-MS/MS platform for 170 chemicals quantification."

Regarding the qualitative accuracy issue of analyte, it is important to emphasize that we used standards for each analyte, 267 standards were used for targeting method establishment following a commonly used protocol based on the triple quadrupole

instrument. We have clarified this in “Chemicals and reagents” section of the manuscript. We used retention times and two ion pairs (quantitative and qualitative) matched to the standard to ensure that the analytes were authentic. It may be a misunderstanding that you mentioned "analytes without standards", because standards are not equivalent to isotope internal standards which is related to the correction process rather than qualitative accuracy. A small number of analytes were not assigned to suitable internal standards in the correction process of multiple internal standards, but they were stable enough by QC evaluation and were therefore also retained in the final method (**Supplementary Tables 6 and 16**). In conclusion, the parameters of each analyte in the process of method establishment were determined according to the corresponding standard, which ensured the qualitative accuracy of the analyte to the greatest extent.

The QQQ-based targeted methods use multi-reaction monitoring (MRM) ion pairs to make quantitation. For the issue of providing MS/MS spectra, we provide the MRM transitions for each analyte in **Supplementary Tables 15 and 16**. For the raw mass spectrometry data, we have referred to some other related articles which did not choose to public individual-level data for the reasons of legal and policy⁴⁻⁶. Therefore, we still hope that the raw data will only be allowed to obtain under supervised conditions. In Table 2, we give the concentration levels of 74 high-frequency exposures in human serum.

3. The description of batch correction approach is vague. First, the authors didn't define what are the batches, are those samples were multiplex bar-coded or simply their data were acquired in a short period of time? I believe the defined batch has no solid foundation. Furthermore, how exactly “batch-specific calibration curve” approach works. The median or mean values of 21 and 24 QCs in GC and LC platform, respectively, were used for standard curve construction? Often studies only use external or internal standards for normalization and batch correction. Please elaborate how the statistical works integrated of standard curve (external standards) and a handful spike-in internal standards. I believe this is the foundation of how accurate the exposome are assessed.

Response: Generally, to ensure the good status of instrument and repeatability and stability of data during long term large-scale sample analysis, routine maintenance of the instrument is very necessary and performed regularly, hence, batches were generated accordingly. In order to provide a clearer description of the concepts of "batch" and "batch-specific calibration curve", **Supplementary Figure 2** has been added to present the real sequence during the instrument analysis. Routine instrument maintenance in this study was mentioned in the “Quality control and assurance.” section, such as blank flushing, ion source cleaning and etc. This process takes around 2 hours, so the time interval between adjacent batches is actually very short. However, in order to minimize the influence of signal drift, we redo the calibration curve at the beginning of each batch. We have also added the definition and schematic diagram of the aforementioned batches to the “Determination of serum chemicals and batches.” section.

Supplementary Figure 2. Running sequence of calibration curves, quality control samples and real samples in instrumental analysis. (A) The composition of each batch in LC-MS/MS platform. (B) The composition of each batch in GC-MS/MS platform.

QC samples is independent of standard curve construction. The entire calibration curve consisted of 18 and 13 concentration points for LC-MS/MS and GC-MS/MS platforms, respectively. The solution of calibration curve was prepared by using matrix-matched mixtures of standards at different concentration gradients. The specific concentrations can be found in the **Supplementary Note 5**. For each analyte, different concentration points were selected to fit the calibration curve based on their respective linear ranges, which can be found in the **Supplementary Tables 5 and 6**. Then, as mentioned before that a calibration curve is run at the beginning of each batch. For the quantitative analysis of samples in the N-th batch, the batch-specific calibration curve quantification was performed according to the calibration curve at the beginning of the N-th batch. This practice helps eliminate errors introduced by pauses in routine instrument maintenance.

Finally, regarding the question of “how the statistical works integrated of and a handful spike-in internal standards”, internal standard calibration curve method was applied for quantification, which is commonly used in precise quantitative research of endogenous metabolites and exogenous chemicals in biological matrices. The point of this method is a group of consistent internal standard (isotopic labeled) mixture was added to both the calibration curve and each sample, and quantification was achieved through regression equations between peak area ratio and concentration. The specific process is as follows: 1) define the appropriate internal standard for each analyte. For the GC platform, internal standard was selected according to similarities in chemical properties and retention behavior, and used to correct chemicals within its neighboring retention time regions. For the LC platform, the internal standards were selected according to the rule of smallest RSD in QCs after internal standard correction (detail see “Quality control and assurance” section). 2) Construct calibration curves after internal standard (IS) correction. Concentrations of analytes in calibration curve (Cc) were set as the independent variable, the ratios between peak areas and corresponding

internal standard are set as the dependent variable ($AREA_{Cc} / AREA_{IS}$). A weighted linear regression model (weight = $1/x$) was used to construct calibration curves. 3) Obtain relative responses of the samples after IS calibration ($AREA_{sample} / AREA_{IS}$). 4) Calculate the concentration of the analyte in samples. Relative responses of samples (step 3) are substituted into the calibration curve in step 2 to obtain the sample concentration.

In addition, to ensure quantitative accuracy, calibration curve was inserted at the beginning of each batch as mentioned before, therefore, the quantitative methods described above were further performed in batch-specific way, means that “each calibration curve was used to quantify the actual samples in its own batch.” This practice helps eliminate errors introduced by pauses in routine instrument maintenance. For the few analytes those not assigned to the internal standard with smallest RSD in QCs, their quantification was achieved by external calibration curve method. The detail of quantitative method has been provided in the **Supplementary Note 6**.

Minor points

1. The title of this article is too broad not very specific addressing the exposome investigation only in east Asia (China), this could lead to race/population bias.

Response: Based on your suggestion, we have changed the title of this article into “An exposome atlas of serum reveals risk of chronic diseases in Chinese population”.

2. In the same regard, this study cannot be called as an exposome atlas with only 267 chemicals.

Response: Although 267 chemicals cannot represent the complete profile of exposome, it is currently the most comprehensive exposome method in existing research. We have acknowledged this limitation in the article, so we would like to maintain the original title here.

3. Figure 1 is overstuffed with many supplemental figures and doesn't contain critical information for understanding the scientific contents, for example, no X and Y axis titles. This is a more like a graph abstract not an actual figure.

Response: You are right. **Figure 1** is more like a graph abstract. Each subgraph is derived from a scaled down version of the actual result graph shown in each section. The purpose of **Figure 1** is to visually present an overview of the entire research process. We believe that **Figure 1** can help readers better understand our research ideas, approaches and the key results presented in each section.

4. In suppl note 4, only LC parameters but no MS ones were addressed, such as cycle time, etc.

Response: We have added MS parameters in suppl note 4. The detailed information is as follows: “In the mass spectrometer system, ionization of targeted analytes was performed by electrospray ionization with positive/negative switching mode. The

electrospray voltage is set at 5500V for positive ion scanning mode and -4500V for negative ion scanning mode. The curtain gas (CUR) is set at 40.0 psi, and the ion source temperature is set at 350 °C. The ion source gases 1 (GS1) and 2 (GS2) are both set at 50 psi. Declustering potential and collision energy voltages were optimized for each chemical based on corresponding standard. Detail information is presented in **Supplementary Table 16.**”

Associated Questions from previous reviewer#1:

o I don't think the authors directly addressed the question “How does it compare to the established literature?”, they simply consolidated a pile of paper from Pubmed (Suppl Table 12), no summary of current knowledge in this field was provided.

Response: In previous responses, we highlighted the significant advantages of this study in terms of chemical coverage and population size, and compared with large number of relevant literatures. Here, we continue to explain the new knowledge provided by this study through a specific comparison with the relevant literatures, and add the related content to the discussion section.

Firstly, this study provides the most representative biomonitoring data for the issue of exposures in the Chinese population. Based on these data, the residual levels of chemicals in serum, their distribution in different regions and age groups, as well as their association with the risk of various chronic diseases have been comprehensively clarified. Compared to other literatures, our study has a larger population scale and more comprehensive exposure analysis.

Secondly, the coverage of chemicals in our study was significantly higher than that in previous studies, which contributed to the new discovery of some high-frequency chemical residues in the blood that have not been previously studied, such as acesulfame, cyclamic acid, fipronil sulfone, and indole-3-butyric acid (IBA). Based on this, we have firstly disclosed the associations between above mentioned chemicals and the risk of chronic diseases. For example, fipronil sulfone is associated with an increased risk of hyperlipidemia, and indole-3-butyric acid is associated with an increased risk of hyperuricemia.

Finally, this study conducted exposure mixtures analysis and identified exposure mixtures related to diseases. Although more and more people recognize the importance of exposure mixtures, there was limited research, especially large-scale population-based studies on exposure mixtures analysis. This study combined three mixture effect models and found that three groups of exposure mixtures showed significant risk-enhancing effects on hyperuricemia, hyperlipidemia, and metabolic syndrome, respectively.

In summary, compared to existing research, this study provides more knowledge and insights in this field.

o I think the authors raised a valid point that triple-quad (QQQ) instrument is lower resolution but higher sensitivity than high-resolution mass spec. The targeted method could be quite useful for future studies. However, the method was not quite clear and

whether it can be adaptable to other QQQ platforms from different vendors.

Response: Thank you for your positive feedback. In order to make the method clearer, we have added additional mass spectrometry parameters, including ion source parameters, declustering potential and collision energy voltages (see **Supplementary Notes 3, 4** and the **Supplementary Tables 15, 16**). Our method can be transferred to other mass spectrometry platforms from different companies after re-optimizing some key parameters. Firstly, the ion source parameters need to be adjusted according to the recommended parameters of different instruments. Additionally, the optimal declustering potential and collision energy voltages for each chemical need to be re-optimized in QQQ instruments produced by different companies, by the way declustering potential voltage has little effect on the intensity. While, the same sample pre-treatment scheme, chromatographic conditions, and MRM transitions can be used.

o As I mentioned in my comments, I believe the results seems to be valid, but the description is not very detailed, please elaborate the batch correction approach. Both Suppl Figure 2 and 5 need clarification. In the Suppl Fig 2, “concentrations of 29 calibration curves” means which data points, the curves consist of several concentration points (low to high), and the internal standards are one chemical or several combined. The suppl Fig. 5 doesn’t show how exactly randomization was done in this study. A tone point in the y axis aligns with multiple runs. I got the point but the figure is not a scientific figure.

Response: Batch correction approaches have been added in **Supplementary Note 6** including the quantification process of the internal standard calibration curve method and the definition of batch-specific quantification.

The calibration curves indeed consisted of several concentration points, and concentrations of low, medium and high are usually selected for accuracy evaluation. In original **Supplementary Figure 2 (Supplementary Figure 3 in the modified version)**, 2, 5, and 20 ng/mL spiked concentrations were selected and used for accuracy evaluation of 29 calibration curves in GC-MS/MS platform. 1, 10, and 100 ng/mL spiked concentrations were selected and used for accuracy evaluation of 20 calibration curves in LC-MS/MS platform. We have added the description in the “Determination of serum chemicals and batches” section, and more detailed values of accuracy are in the **Supplementary Tables 3 and 4**.

For internal standard correction process in this study, only one internal standard was assigned to each analyte rather than several combined internal standards. It is probable that the description of “multiple internal standards correction” confused your understanding. Here, multiple internal standards mean those 27 internal standards spiked in total.

Supplementary Figure 5 was intended to show that the control samples and the chronic disease samples were randomly arranged during the instrumental analysis. We agree that this figure doesn’t exactly represent the randomization process, so we have replaced it with a tabular format (**Supplementary Table 7**).

o How to determine the analyte without IS for every analyte using a low-res MS, both retention time and MS/MS transitions have to be matched as in co-eluted peaks (light and heavy chemicals). Authors need to elaborate.

Response: As mentioned in the response to “Major points 2”, there is a corresponding standard for each analyte. This target method was established following a commonly used protocol based on the triple quadrupole instrument. Both retention time and 2 MRM transitions were matched with the corresponding standard. This process is independent of internal standards (isotope standard substances). We have clarified this in “Chemicals and reagents” section of the manuscript.

o I don't think enough info has been provided even in this version, more details of statistical analysis are needed, and possible to deposit the actual method file to supplemental material or online repositories with the MS raw files.

Response: As mentioned above, we have made R code and example files for statistical analysis available on GitHub (see <https://github.com/youlei2023/ExposomeAtlas>). The actual method files have been deposited to supplemental material.

o As I mentioned above, Suppl Fig 5 is not very specific about the random order, the original sequence file can be alternative as a new supplemental table.

Response: We have deleted the **Supplementary Figure 5** and replaced it with **Supplementary Table 7**. We have also modified corresponding part in the manuscript.

1. Zhao F, Wan Y, Zhao H, Hu W, Mu D, Webster TF, et al. Levels of Blood Organophosphorus Flame Retardants and Association with Changes in Human Sphingolipid Homeostasis. *Environ Sci Technol*. 2016; 50: 8896-903.
2. Wang J, Pan Y, Cui Q, Yao B, Wang J, Dai J. Penetration of PFASs Across the Blood Cerebrospinal Fluid Barrier and Its Determinants in Humans. *Environ Sci Technol*. 2018; 52: 13553-61.
3. Zang L, Liu X, Xie X, Zhou X, Pan Y, Dai J. Exposure to per- and polyfluoroalkyl substances in early pregnancy, risk of gestational diabetes mellitus, potential pathways, and influencing factors in pregnant women: A nested case-control study. *Environ Pollut*. 2023; 326: 121504.
4. Chen DQ, Cao G, Chen H, Argyopoulos CP, Yu H, Su W, et al. Identification of serum metabolites associating with chronic kidney disease progression and anti-fibrotic effect of 5-methoxytryptophan. *Nat Commun*. 2019; 10: 1476.
5. Watanabe K, Wilmanski T, Diener C, Earls JC, Zimmer A, Lincoln B, et al. Multiomic signatures of body mass index identify heterogeneous health phenotypes and responses to a lifestyle intervention. *Nat Med*. 2023; 29: 996-1008.
6. Christiansen C, Tomlinson M, Eliot M, Nilsson E, Costeira R, Xia Y, et al. Adipose methylome integrative-omic analyses reveal genetic and dietary metabolic health drivers and insulin resistance classifiers. *Genome Medicine*. 2022; 14.

REVIEWERS' COMMENTS

Reviewer #3 (Remarks to the Author):

No further comments.

Reviewer #4 (Remarks to the Author):

I believe the authors have thoroughly and adequately addressed all my comments.